# A composite double-/single-stranded RNA-binding region in protein Prp3 supports tri-snRNP stability and splicing

Sunbin Liu[1], Sina Mozaffari-Jovin[2], Jan Wollenhaupt[1], Karine F Santos[1], Matthias Theuser[1], Stanislaw Dunin-Horkawicz[3], Patrizia Fabrizio[2], Janusz M Bujnicki[3,4], Reinhard Lührmann[2]*, Markus C Wahl[1]*

[1]Laboratory of Structural Biochemistry, Freie Universität Berlin, Berlin, Germany; [2]Department of Cellular Biochemistry, Max Planck Institute for Biophysical Chemistry, Göttingen, Germany; [3]Laboratory of Bioinformatics and Protein Engineering, International Institute of Molecular and Cell Biology, Warsaw, Poland; [4]Institute of Molecular Biology and Biotechnology, Faculty of Biology, Adam Mickiewicz University, Poznan, Poland

**Abstract** Prp3 is an essential U4/U6 di-snRNP-associated protein whose functions and molecular mechanisms in pre-mRNA splicing are presently poorly understood. We show by structural and biochemical analyses that Prp3 contains a bipartite U4/U6 di-snRNA-binding region comprising an expanded ferredoxin-like fold, which recognizes a 3′-overhang of U6 snRNA, and a preceding peptide, which binds U4/U6 stem II. Phylogenetic analyses revealed that the single-stranded RNA-binding domain is exclusively found in Prp3 orthologs, thus qualifying as a spliceosome-specific RNA interaction module. The composite double-stranded/single-stranded RNA-binding region assembles cooperatively with Snu13 and Prp31 on U4/U6 di-snRNAs and inhibits Brr2-mediated U4/U6 di-snRNA unwinding in vitro. RNP-disrupting mutations in Prp3 lead to U4/U6•U5 tri-snRNP assembly and splicing defects in vivo. Our results reveal how Prp3 acts as an important bridge between U4/U6 and U5 in the tri-snRNP and comparison with a Prp24-U6 snRNA recycling complex suggests how Prp3 may be involved in U4/U6 reassembly after splicing.

*For correspondence: Reinhard. Luehrmann@mpi-bpc.mpg.de (RL); mwahl@zedat.fu-berlin. de (MCW)

Competing interests: The authors declare that no competing interests exist.

## Introduction

Pre-mRNA splicing is catalyzed by a multi-subunit RNA-protein (RNP) enzyme, the spliceosome, which facilitates two successive *trans*esterification reactions (steps 1 and 2) that lead to the removal of an intron and the ligation of its flanking exons. For each splicing event, a spliceosome is newly formed via the stepwise recruitment of small nuclear (sn) RNPs (U1, U2, U4, U5 and U6 in the case of the major spliceosome) and numerous non-snRNP proteins to a pre-mRNA substrate (*Wahl et al., 2009*; *Will and Lührmann, 2011*). During canonical cross-intron spliceosome assembly, U1 and U2 snRNPs bind the 5′-splice site (5′SS) and the branch point region of an intron, respectively (*Mount et al., 1983*; *Kramer et al., 1984*; *Parker et al., 1987*; *Wu and Manley, 1989*; *Zhuang and Weiner, 1989*). Formation of this A complex is followed by the recruitment of a preformed U4/U6•U5 tri-snRNP, yielding the pre-catalytic B complex (*Bindereif and Green, 1987*; *Cheng and Abelson, 1987*; *Konarska and Sharp, 1987*; *Deckert et al., 2006*; *Fabrizio et al., 2009*), which requires extensive conformational and compositional rearrangements to form a catalytically active spliceosome. Catalytic activation includes the disruption of the U1/5′SS interaction (*Konforti et al., 1993*) and the separation of U4 snRNA from U6 snRNA (*Konarska and Sharp, 1987*; *Yean and Lin, 1991*), which are extensively base-paired in the tri-snRNP and in the B complex, leading to the dissociation of U1 snRNP, U4 snRNA

**eLife digest** Proteins are built following instructions contained within the DNA of gene sequences. This genetic information is copied into short-lived molecules, called messenger RNAs (or mRNAs), which move away from the DNA and are then decoded by the molecular machines that build proteins. However, mRNA sequences often have to be edited before they are used. Another molecular machine, called a spliceosome, carries out some of this editing.

A spliceosome is formed from a number of smaller subunits, including three RNA-protein particles that each contain one RNA molecule (called U1, U2 and U5), and one particle that contains two RNA molecules (called U4 and U6). These subunits must assemble around an unedited mRNA in a particular order so that the spliceosome can work correctly. Once the mRNA has been edited, and the spliceosome has performed its job, these complexes need to disassemble so that they are ready to be reassembled around a new mRNA molecule. A protein called Prp3 is known to be involved in these assembly, disassembly and reassembly steps. However, it is unclear how this protein performs these activities.

Liu et al. have now used structural biology and biochemical techniques to determine the three-dimensional structure of Prp3, and have shown that this protein has a "two-part" binding site that binds to the RNA molecules in the U4/U6 subunit of the spliceosome. Further analyses revealed that one of these features is only found in Prp3 and not in other types of RNA-binding proteins.

Together with previous work, Liu et al. also reveal that Prp3 can serve as a 'bridge' between the U4/U6 and U5 subunits of the spliceosome, and suggest how these features allow the two subunits to group together before they are incorporated into a spliceosome.

Notably, certain mutations in the gene for the Prp3 protein lead to a human eye disease called retinitis pigmentosa. In the future it will be important to investigate if the above activities are affected in the mutant variants of the Prp3 protein.

and all U4/U6-associated proteins. These rearrangements give rise to the $B^{act}$ complex (*Fabrizio et al., 2009*; *Bessonov et al., 2010*) and, upon further remodeling, to the B* complex (*Warkocki et al., 2009*), in which U6 snRNA forms a central part of the spliceosome's active site. The B* complex facilitates the first step of splicing, the ensuing C complex (*Konarska et al., 2006*; *Bessonov et al., 2008*; *Fabrizio et al., 2009*) catalyzes the second step of splicing, after which the spliceosome releases its products and the remaining snRNPs and non-snRNP factors are recycled.

To participate in further rounds of splicing, the U4/U6•U5 tri-snRNP must be reassembled by initial dimerization of U4 and U6 snRNPs followed by association with U5 snRNP (*Stanek and Neugebauer, 2006*). Association of U4 and U6 snRNPs is mediated in part by base pairing between their respective snRNAs, which form two inter-molecular helices (stems I and II) that are separated by a U4 5′-stem-loop (5′SL; *Figure 1A*). U4/U6 base pairing is mutually exclusive with the U6 snRNA conformation in the activated spliceosome. Reannealing of U4 and U6 snRNAs after splicing thus requires the Prp24 assembly chaperone in yeast (*Raghunathan and Guthrie, 1998*) or its SART3 ortholog in human (*Bell et al., 2002*), which transiently bind U6 snRNA, as well as the LSm proteins (*Achsel et al., 1999*; *Rader and Guthrie, 2002*; *Verdone et al., 2004*), which bind and remain at the 3′-end of U6 snRNA (*Beggs, 2005*; *Zhou et al., 2014*). In addition, the U4-specific Prp3 protein is required for U4/U6 di-snRNP and U4/U6•U5 tri-snRNP formation (*Anthony et al., 1997*) but molecular mechanisms underlying its functions are poorly understood.

Human (h) and yeast (y) Prp3 form a complex with the respective Prp4 proteins (*Ayadi et al., 1998*; *Gonzalez-Santos et al., 2002*). hPrp3 can be crosslinked to the U6 snRNA portion of a U4/U6 di-snRNA complex comprising the U4 5′SL, an intact stem II and a U6 3′-overhang (*Nottrott et al., 2002*) and can pull down U4 and U6 snRNAs from nuclear extract (*Gonzalez-Santos et al., 2002*), pointing towards direct Prp3-snRNA interactions. hPrp3 also interacts with the U5-specific proteins hPrp6 and hSnu66 (*Liu et al., 2006*).

Except for an N-terminal region, Prp3 is highly conserved from yeast to human and contains a C-terminal domain of unknown function (DUF1115; PFAM ID PF06544; *Figure 1A*; *Figure 1—figure supplement 1*). Recent homology modeling has predicted a ferredoxin-like fold for the human Prp3 DUF1115 domain (*Korneta et al., 2012*). Here, we demonstrate that C-terminal regions of Prp3,

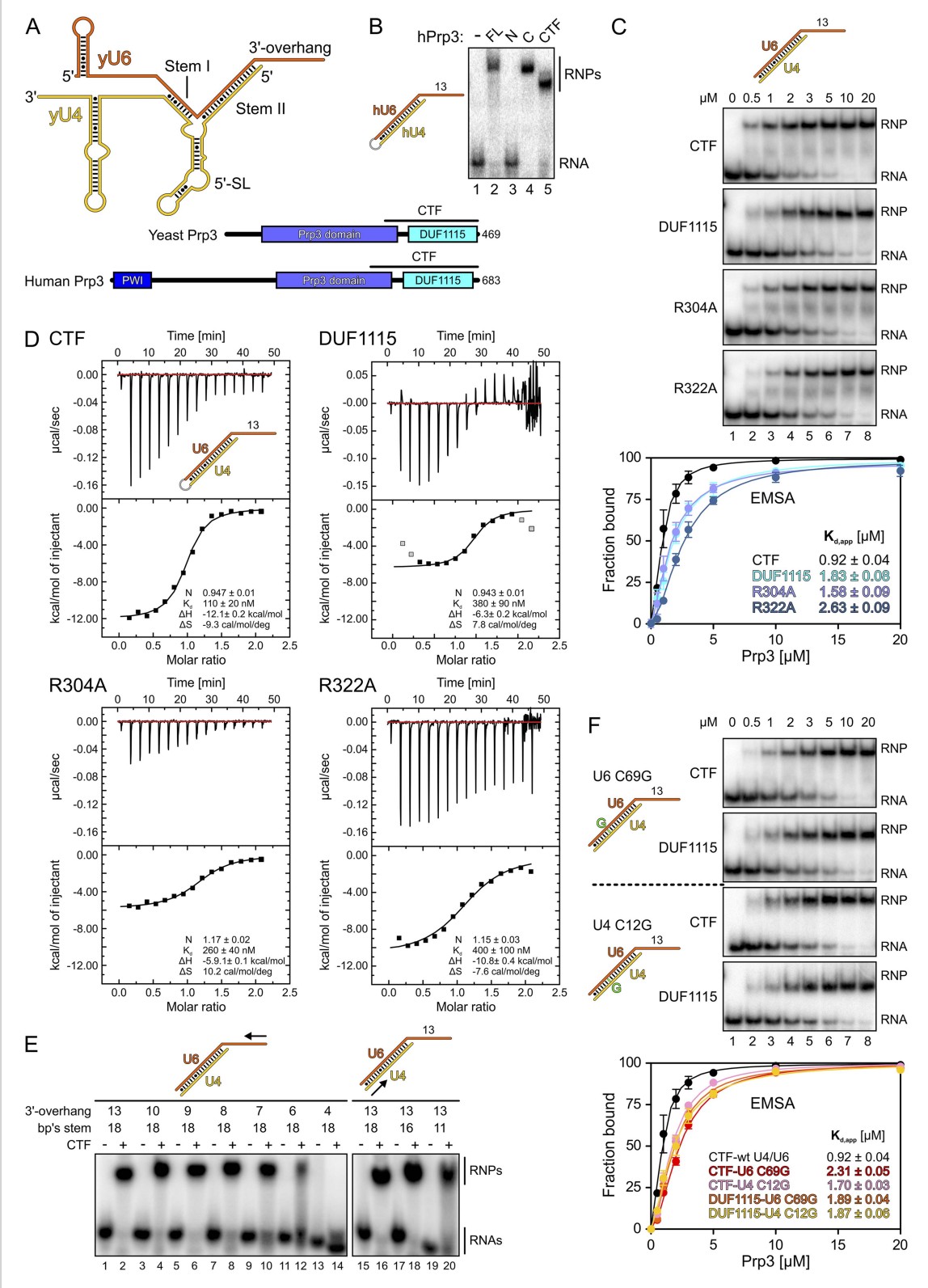

**Figure 1**. Protein and RNA requirements for Prp3 binding to U4/U6 di-snRNA. (**A**) Schematic presentation of yeast U4/U6 di-snRNA (yU4—gold; yU6—orange) and domain organizations of yeast and human Prp3. Regions corresponding to the C-terminal U4/U6-binding fragments (CTF) of the proteins are indicated by black lines above the schemes. (**B**) ESMA monitoring binding of hPrp3 protein variants (as MBP or GST fusions; 25 μM) to hU4/U6^stem II+13nt (scheme on the left). hPrp3 constructs are indicated above the lanes. FL—full-length; N—residues 1–442; C—residues 195–683; CTF—residues

*Figure 1. continued on next page*

Figure 1. Continued

484–683. Bands are identified on the right; RNA—unbound RNA; RNPs—RNA–protein complexes. (**C**) EMSA titrations monitoring binding of yPrp3$^{CTF}$ variants and yPrp3$^{DUF1115}$ (proteins indicated at the left of the gels) to yU4/U6$^{stem\ II+13nt}$ (scheme on the top). Protein concentrations in each lane are indicated above the first gel. Bottom: quantification of the data above. The data were fit to a single exponential Hill equation (fraction bound = A [protein]$^n$/([protein]$^n$ + K$_d$$^n$): A, fit maximum of RNA bound; n, Hill coefficient) (*Ryder et al., 2008*). Errors indicate standard errors of the mean of at least two independent experiments. (**D**) Isothermal titration calorimetry monitoring interactions between yPrp3$^{CTF}$ variants or yPrp3$^{DUF1115}$ (proteins indicated above and on the left of each panel) to yU4/U6$^{stem\ II+13nt}$ (scheme in the first panel). The proteins, in particular yPrp3$^{DUF1115}$ and yPrp3$^{CTF,R322A}$, tended to aggregate when added in excess of available RNA binding sites, giving rise to background signals at the ends of some runs. Data points in gray in the DUF1115 analysis were omitted during the fitting. Deduced binding stoichiometries (N), K$_d$'s, enthalpies (ΔH) and entropies (ΔS) of the interactions are listed in the lower parts of the panels. (**E**) Binding of yPrp3$^{CTF}$ (20 µM) to the indicated fragments of yU4/U6 (schemes above the gels; arrows indicate sequential shortening of RNA elements). Lanes 1–14—shortening of the yU6 3'-overhang. Lanes 15–20—shortening of stem II. (**F**) EMSA titrations monitoring binding of binding of yPrp3$^{CTF}$ or yPrp3$^{DUF1115}$ to yU4/U6$^{stem\ II+13nt}$ bearing yU6 C69 G (top two gels) or yU4 C12 G (bottom two gels) exchanges that restore Watson–Crick base pairing (schemes on the far left; mutant residues highlighted in green). Proteins are indicated at the left of the gels. Protein concentrations in each lane are indicated above the first gel. Bottom: quantification as in panel 1C. CTF-wt U4/U6—reference copied from panel 1C. Errors indicate standard errors of the mean of at least two independent experiments.

The following figure supplements are available for figure 1:

**Figure supplement 1**. Protein sequence comparisons.

**Figure supplement 2**. RNA sequence comparisons.

including DUF1115, bind U4/U6 di-snRNA fragments containing stem II and a U6 3'-overhang and elucidated crystal structures of a yPrp3 C-terminal region alone and in complex with a U4/U6 di-snRNA fragment. Structure-guided mutations that led to reduced U4/U6 interaction in vitro and to reduced cell viability also reduced U4/U6•U5 tri-snRNP levels and splicing in vivo. Our results indicate how Prp3 functionally bridges U4/U6 and U5 in the tri-snRNP by Prp3-RNA interactions on one side and Prp3-protein interactions on the other. Moreover, a comparison with the structure of a Prp24-U6 snRNA complex (*Montemayor et al., 2014*) suggests how Prp3 may initiate the handover of U6 snRNA from the Prp24 recycling factor to U4 snRNP during U4/U6 reassembly.

## Results

### Characterization of a conserved C-terminal U4/U6 di-snRNA-binding region in Prp3

Previous studies have shown that human (h) Prp3 in the context of a hPrp3-hPrp4-hCypH complex contacts U6 snRNA in a region that forms stem II and a U6 single-stranded 3'-overhang (*Nottrott et al., 2002*). To test whether hPrp3 alone is sufficient for stable RNA binding and to delineate hPrp3 elements required for complex formation, we produced full-length hPrp3 (hPrp3$^{FL}$) and fragments lacking ca. 200–250 residues from either end (hPrp3$^N$—residues 1–442; hPrp3$^C$—residues 195–683) as N-terminal maltose-binding protein [MBP] fusion proteins. We then tested binding of these proteins to a hU4/U6 construct containing stem II (fused by a GAAA tetraloop) and a 13 nucleotide [nt] U6 3'-overhang (hU4/U6$^{stem\ II+13nt}$) in electrophoretic mobility shift assays (EMSAs). Only the full-length protein and hPrp3$^C$, but not hPrp3$^N$, bound hU4/U6$^{stem\ II+13nt}$ (*Figure 1B*, lanes 2–4). Trypsin treatment of the hPrp3$^C$-hU4/U6$^{stem\ II+13nt}$ complex gave rise to a protein fragment containing residues 484–683 (hPrp3$^{CTF}$) as shown by mass spectrometric fingerprinting and N-terminal sequencing. hPrp3$^{CTF}$ contains the predicted DUF1115 domain (residues 540–683) and a conserved, preceding peptide rich in basic amino acid residues (*Figure 1A*; *Figure 1—figure supplement 1*). Binding of recombinant hPrp3$^{CTF}$ (as a glutathione S-transferase [GST] fusion) to hU4/U6$^{stem\ II+13nt}$ was comparable to hPrp3$^{FL}$ or hPrp3$^C$ (*Figure 1B*, lane 5). These results show that the C-terminal ca. 200 amino acids of hPrp3 encompass the protein elements that mediate stable U4/U6 binding.

All following experiments were performed with yeast (y) factors. To test whether the C-terminal U4/U6 di-snRNA-binding region is conserved in yPrp3, we produced a protein comprising the 177 C-terminal residues of yPrp3 (residue 296–469; yPrp3$^{CTF}$). yPrp3$^{CTF}$ bound a yU4/U6 duplex containing the complete stem II and a 13-nt yU6 3'-overhang (yU4/U6$^{stem\ II+13nt}$) with an apparent

dissociation constant ($K_{d,app}$) of 0.92 µM as determined by EMSA (*Figure 1C*, first gel and quantification) and with a $K_d$ of 110 nM as determined by isothermal titration calorimetry (ITC; *Figure 1D*, top left). The lower apparent affinity estimated by EMSA is likely due to the presence of non-specific tRNA competitor in this assay. The yPrp3 DUF1115 domain (residues 325–469; yPrp3$^{DUF1115}$) lacking the preceding basic peptide showed ca. twofold reduced affinity for yU4/U6$^{stem\ II+13nt}$ in EMSA ($K_{d,app}$ 1.83 µM; *Figure 1C*, second gel and quantification) and a ca. 3.5-fold lower affinity in ITC ($K_d$ 380 µM; *Figure 1D*, top right). Exchange of conserved arginine residues at positions 304 and 322 in the N-terminal basic peptide of yPrp3$^{CTF}$ (*Figure 1—figure supplement 1*), which could directly interact with the negatively charged RNA backbone, reduced affinities for yU4/U6$^{stem\ II+13nt}$ in both EMSA ($K_{d,app}$ 1.58 µM and 2.63 µM for yPrp3$^{CTF,R304A}$ and yPrp3$^{CTF,R322A}$, respectively; *Figure 1C*, third and fourth gels and quantification) and ITC ($K_d$ 260 nM and 400 nM for yPrp3$^{CTF,R304A}$ and yPrp3$^{CTF,R322A}$, respectively; *Figure 1D*, bottom left and right) to a similar extent as removal of the entire preceding peptide. Furthermore, the thermodynamic signatures of the interactions involving yPrp3$^{DUF1115}$, yPrp3$^{CTF,R304A}$ and yPrp3$^{CTF,R322A}$ changed considerably compared to yPrp3$^{CTF}$. The DUF1115 domain as well as both arginine-to-alanine variants (in particular yPrp3$^{CTF,R304A}$) exhibited significantly less favorable interaction enthalpies and more favorable (or less unfavorable) interaction entropies compared to yPrp3$^{CTF}$ (*Figure 1D*). One explanation for these observations could be that the N-terminal peptide is a flexible element in yPrp3$^{CTF}$, which becomes immobilized (loss in conformational entropy) by RNA contacts (gain in interaction enthalpy) upon binding of yU4/U6$^{stem\ II+13nt}$. In any case, these observations show that both the DUF1115 domain and the preceding basic peptide contribute to the RNA binding.

Next, we further probed the RNA requirements for stable binding. yPrp3$^{CTF}$ efficiently bound a yU4/U6 constructs bearing full-length stem II and yU6 3′-overhangs of at least eight nts (*Figure 1E*, lanes 1–8), while further shortening of the yU6 3′-overhang led to progressively reduced binding (lanes 9–14). Removal of the first two A-U base pairs from the 5′-end of stem II had no consequence for binding of yPrp3$^{CTF}$ (*Figure 1E*, lanes 15–18), but reduced binding was seen when seven base pairs, including a non-canonical C69$^{U6}$-C12$^{U4}$ pair, were removed (*Figure 1E*, lane 20). A non-canonical pyrimidine–pyrimidine base pair is conserved in the corresponding stem II regions of U4/U6 from other organisms (*Figure 1—figure supplement 2*). When we converted yU6 C69 or yU4 C12 of the non-canonical C-C pair in stem II to G's to allow Watson-Crick base pairing at these positions, the affinity of yPrp3$^{CTF}$ was reduced about twofold in EMSA titrations (*Figure 1F*, first and third gels and quantification). In contrast, the DUF1115 domain alone did not exhibit reduced affinity to the mutant RNAs compared to the wt (*Figure 1F*, second and fourth gels and quantification). These findings suggest that yPrp3$^{CTF}$ recognizes portions of the U6 3′-overhang as well as parts of yU4/U6 stem II, including the non-canonical C69$^{U6}$-C12$^{U4}$ pair.

Binding of Prp3 to U4/U6 stem II would be expected to stabilize the U4/U6 duplex, which is unwound by the Brr2 helicase during spliceosome catalytic activation. To test relative contributions of the yPrp3 DUF1115 domain and the preceding basic peptide to stabilization of yU4/U6, we therefore assessed the influence of yPrp3$^{CTF}$ and yPrp3$^{DUF1115}$ on Brr2-mediated U4/U6 unwinding. Addition of yPrp3$^{CTF}$ reduced the rate of yBrr2-mediated yU4/U6 unwinding about fivefold, while yPrp3$^{DUF1115}$ showed almost no effect (ca. 1.2-fold reduction; *Figure 2A,B*).

Together, these analyses show that Prp3 orthologs contain a conserved, C-terminal region (Prp3$^{CTF}$) that is necessary and sufficient for stable binding to U4/U6 di-snRNAs and that upper parts of stem II and at least eight nts of the U6 3′-overhang are required for stable binding. Prp3$^{CTF}$ comprises the DUF1115 domain and a preceding basic peptide, both of which contribute to RNA binding.

## Cross talk of U4/U6-bound proteins

Previous assembly and structural studies had shown that the U4/U6-specific proteins Snu13, Prp31 and Prp3 bind U4/U6 di-snRNAs in a cooperative manner (*Nottrott et al., 2002*; *Liu et al., 2007*). To see whether the C-terminal U4/U6-binding region of Prp3 is sufficient to sense pre-bound Snu13 and Prp31 proteins, we compared binding of yPrp3$^{CTF}$ and yPrp3$^{DUF1115}$ to a fused yU4/U6-like RNA (yU4/U6$^{fused}$; *Figure 2C*) alone or pre-bound to ySnu13 or ySnu13 and yPrp31$^{1–462}$ under identical conditions. Both yPrp3 variants bound the ySnu13-yPrp31$^{1–462}$-yU4/U6$^{fused}$ complex (*Figure 2C*, lanes 13–18, right) more efficiently than the ySnu13-yU4/U6$^{fused}$ complex (lanes 7–12, middle), which in turn was bound better than the naked RNA (lanes 1–6, left). However, yPrp3$^{CTF}$ interacted more stably than yPrp3$^{DUF1115}$ with the naked RNA (lanes 4–6 vs lanes 1–3) and with the ySnu13-yU4/U6$^{fused}$ complex

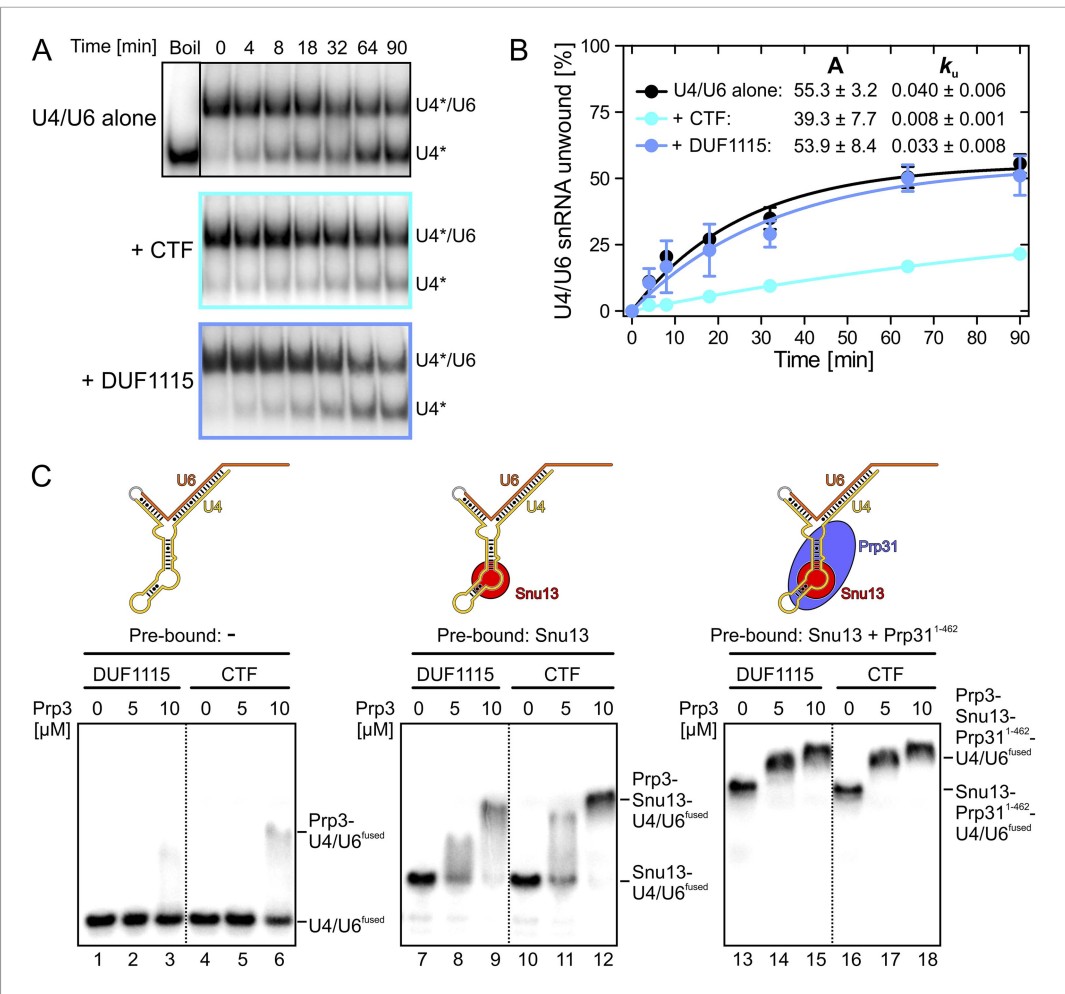

**Figure 2**. Effects on Brr2-mediated U4/U6 unwinding and interplay with other U4/U6 proteins. (**A**) Native gels monitoring yU4/U6 di-snRNA unwinding by yBrr2 in the absence of other proteins (top), in the presence of yPrp3[CTF] (middle) or yPrp3[DUF1115] (bottom). Asterisks indicate radioactive label on yU4 snRNA. (**B**) Quantification of the data in (**A**). The data were fit to a single exponential equation: % duplex unwound = A{1 − exp(−$k_u$ t)}; A—amplitude of the reaction; $k_u$—apparent first-order rate constant; t—time. Amplitudes and rate constants are listed. Errors indicate standard errors of the mean of four independent experiments. (**C**) Binding of yPrp3[DUF1115] (left three lanes of each panel) or yPrp3[CTF] (right three lanes of each panel) to yU4/U6[fused] (lanes 1–6; left), yU4/U6[fused] pre-bound to Snu13 (lanes 7–12; middle) or yU4/U6[fused] pre-bound to Snu13 and Prp31[1−462] (lanes 13–18; right) under otherwise identical conditions. All three panels are from the same gel and were regrouped. Schemes of yU4/U6[fused] alone or pre-bound by proteins are shown on the top.

(lanes 10–12 vs lanes 7–9). The sensitivity of the assay was insufficient to resolve possible binding differences to the ySnu13-yPrp31[1−462]-yU4/U6[fused] complex (lanes 13–18). These results show that yPrp3[CTF] binds cooperatively with ySnu13 and yPrp31 to yU4/U6, in part based on the basic peptide preceding the DUF1115 domain.

## Crystal structure of the C-terminal yU4/U6-binding region of yPrp3

Sequence analyses of the C-terminal U4/U6-binding portions of Prp3 proteins did not reveal any obvious similarities to known RNA-binding domains. We therefore determined the crystal structure of yPrp3[CTF] at 2.7 Å resolution (*Table 1*). The protein crystallized with three monomers per asymmetric unit (*Figure 3—figure supplement 1A*). The electron density allowed modeling of residues 335–467, corresponding to the predicted DUF1115 domain. Apart from the C-terminal

**Table 1.** Crystallographic data

| Data set | yPrp3$^{296-469}$ | yPrp3$^{325-469}$ | yPrp3$^{296-469}$-yU4/U6$^{stem\ II-2}$ |
|---|---|---|---|
| Data collection | | | |
| Wavelength (Å) | 0.91840 | 0.91841 | 0.97968 |
| Space group | C222$_1$ | P6$_5$ | C2 |
| Unit cell parameters | | | |
| a, b, c (Å) | 87.7, 161.2, 105.2 | 56.1, 56.1, 86.8 | 144.7, 59.6, 109.8 |
| α, β, γ (°) | 90.0, 90.0, 90.0 | 90.0, 90.0, 120.0 | 90.0, 118.5, 90.0 |
| Resolution (Å) | 50.0–2.70 (2.80–2.70)* | 50.0–2.00 (2.12–2.00) | 30.0–3.25 (3.33–3.25) |
| Reflections | | | |
| Unique | 20,036 (1993) | 10,459 (1622) | 24,895 (1844) |
| Completeness (%) | 97.3 (99.0) | 99.3 (96.2) | 97.9 (98.5) |
| Redundancy | 3.4 (3.3) | 11.4 (11.3) | 2.0 (1.9) |
| R$_{meas}$† | 0.066 (0.791) | 0.072 (0.459) | 0.017 (0.146) |
| I/σ (I) | 13.31 (1.11) | 24.96 (5.94) | 5.38 (0.90) |
| CC (1/2)‡ | – | 99.9 (95.5) | 99.4 (42.0) |
| Refinement | | | |
| Resolution (Å) | 30.00–2.70 (2.77–2.70) | 24.28-2.00 (2.20–2.00) | 29.78-3.25 (3.38–3.25) |
| Reflections | | | |
| Number | 18,932 (1146) | 10,454 (2424) | 24,894 (2631) |
| Completeness (%) | 95.8 (80.5) | 99.3 (97.0) | 98.3 (98.0) |
| Test set (%) | 5.2 | 5.0 | 5.0 |
| R factors§ | | | |
| R$_{work}$ (%) | 20.7 (37.0) | 16.1 (16.6) | 24.9 (36.9) |
| R$_{free}$ (%) | 26.2 (42.7) | 22.1 (21.9) | 29.9 (38.9) |
| RMSD# | | | |
| Bond lengths (Å) | 0.010 | 0.013 | 0.007 |
| Bond angles (°) | 1.316 | 1.278 | 1.006 |
| Ramachandran plot¶ (%) | | | |
| Favored | 96.57 | 98.56 | 90.87 |
| Allowed | 2.64 | 1.44 | 7.98 |
| Outlier | 0.79 | 0.00 | 1.14 |

*Values for the highest resolution shell in parentheses.

†R$_{meas}$ = Σ$_h$[n/(n − 1)]$^{1/2}$Σ$_i$|I$_h$ − I$_{h,i}$|/Σ$_h$Σ$_i$I$_{h,I}$, where $I_h$ is the mean intensity of symmetry-equivalent reflections and $n$ is the redundancy.

‡CC (1/2) is the percentage of correlation between intensities from random half-data sets.

§R = Σ$_{hkl}$||F$_{obs}$| − |F$_{calc}$||/Σ$_{hkl}$|F$_{obs}$|; R$_{work}$ − hkl ∉ T; R$_{free}$ − hkl ∈ T; T—test set.

#Root-mean-square deviation from target geometries.

¶Calculated with MolProbity (http://molprobity.biochem.duke.edu/).

two residues, the N-terminal 39 residues (296–334; comprising the preceding basic peptide) could not be traced, indicating that they indeed constitute an intrinsically disordered or flexibly attached element, as surmised based on the ITC experiments.

The core of the yPrp3 DUF1115 domain resembles the homology model of the corresponding human domain (*Korneta et al., 2012*) (root-mean-square deviation [RMSD] of 2 Å for 87 common Cα atoms). The structure exhibits a five-stranded mixed β-sheet with two α-helices (α1 and α2) running parallel to the β-strands on one side of the sheet and one (α3) on the other (*Figure 3A*). The first 98 residues of the structure (residues 335–432) form a α/β sandwich with a ferredoxin-like topology and

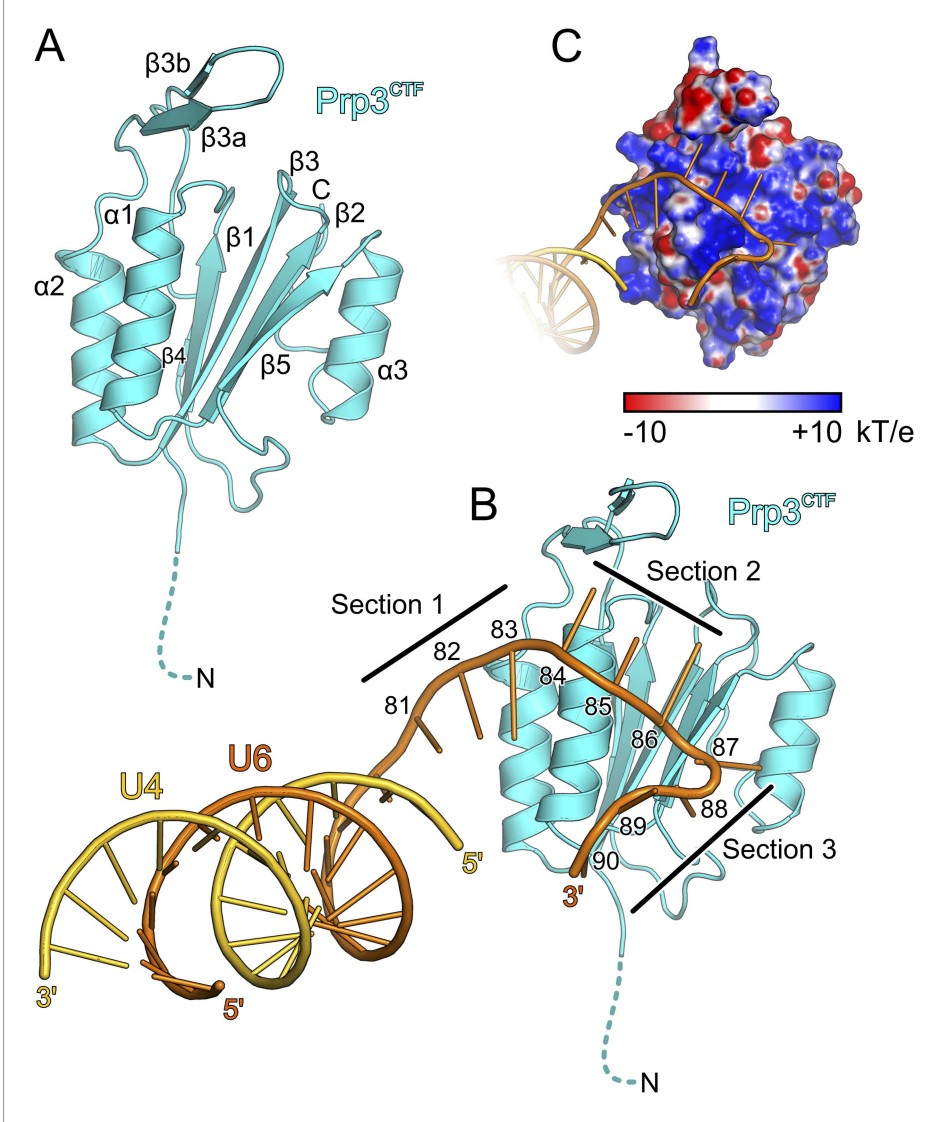

**Figure 3**. Structural overviews. (**A**) Structure of a yPrp3$^{CTF}$. Secondary structure elements and termini are labeled. Dashed line indicates residues contained in the construct but not visible in the electron density. (**B**) Structure of a yPrp3$^{CTF}$-yU4/U6$^{stem\ II+10nt}$ complex with the protein in the same orientation as in *Figure 3A*. yPrp3$^{CTF}$—cyan; yU4—gold; yU6—orange. Dashed line indicates residues contained in the construct but not visible in the electron density. Sections 1–3, between which the yU6 3′-overhang changes direction on the surface of yPrp3$^{CTF}$, are indicated by black lines. Residues in the yU6 3′-overhang are numbered. (**C**) Electrostatic surface potential of yPrp3$^{CTF}$ in complex with yU4/U6$^{stem\ II+10nt}$. Blue—positive; red—negative. Units kT/e with k—Boltzmann''s constant, T—absolute temperature, E—charge of an electron.

The following figure supplements are available for figure 3:

**Figure supplement 1**. Structural comparisons.

**Figure supplement 2**. Phylogenetic analysis.

an additional, long β-hairpin (residues 403–421, comprising strands β3a and β3b) inserted between helix α2 and strand β4. The ferredoxin-like fold is further expanded by an extra β-strand (β5), a helix (α3) and a following loop at the C-terminus. In two of the three independent molecules, the β3a/β3b hairpins adopt very similar conformations with their long axes oriented perpendicular to the direction

of the strands in the central β-sheet. Residues D405 and D418 at the bases of the hairpins coordinate an $Yt^{3+}$ ion. The third molecule lacks electron density for large parts of the hairpin (residues 407–421), suggesting that crystal packing and a bound metal ion may have stabilized this normally flexible element in the first two molecules.

We also determined the crystal structure of yPrp3$^{DUF1115}$, lacking the preceding basic peptide, at 2.0 Å resolution (*Table 1*). While the ordered parts of yPrp3$^{CTF}$ and yPrp3$^{DUF1115}$ are very similar (RMSD of 0.48 Å for 131 common Cα atoms; *Figure 3—figure supplement 1A*), the β3a/β3b hairpin in yPrp3$^{DUF1115}$ is positioned closer to the globular part of the protein, showing that its relative positioning is indeed flexible.

## Crystal structure of a yPrp3$^{CTF}$-yU4/U6$^{stem\ II+10nt}$ complex

To elucidate the molecular mechanism underlying U4/U6 di-snRNA binding by yPrp3$^{CTF}$, we determined its crystal structure in complex with yU4/U6$^{stem\ II+10nt}$ at 3.25 Å resolution (*Table 1*; *Figure 3B*). Residues 335–468 of yPrp3$^{CTF}$ and all nts (G81-U90) of the yU6 3′-overhang were well ordered in the two complexes contained in an asymmetric unit. The electron density for the yU4/U6 stem II regions was less well defined, in particular in the part of stem II distal to the duplex-to-single strand junction in one of the complexes. The electron density was consistent with the stem II regions adopting standard A-form geometry in both complexes but did not allow us to model in detail possible local deviations, for example, around a non-canonical C69$^{U6}$-C12$^{U4}$ base pair.

Protein–RNA interactions in the well-defined portions of the two crystallographically independent complexes were largely identical (*Figure 3—figure supplement 1B*). In both complexes, the single-stranded (ss) yU6 3′-overhang arches across helix α1 and strand β5 of yPrp3$^{CTF}$, running below the long β3a/β3b hairpin (*Figure 3B*). Its binding surface on the protein is carpeted by an electropositive surface potential (*Figure 3C*). The yU6 3′-overhang can be divided into three sections, between which its backbone changes directions on the surface of yPrp3$^{CTF}$ (*Figure 3B*). Nts G81-A83 (section 1) stack on the terminal U80$^{U6}$-A1$^{U4}$ base pair of stem II and run diagonally from the N-terminus of helix α2 to the N-terminus of helix α1; nts C84-G86 (section 2) are positioned perpendicular across the N-terminal end of helix α1; nts U87-U90 (section 3) run along the exposed edge of strand β5 and the C-terminus of helix α1. As a consequence, the end of the U6 3′-overhang is directed via two ca. 90 ° kinks back towards stem II. A RNA structural similarity search using the RNA Bricks database (*Chojnowski et al., 2014*) failed to uncover a case of a similar RNA redirection on the surface of a single protein domain. Notably, the unusual doubly-kinked RNA conformation is stabilized by protein elements that expand the core ferredoxin fold in yPrp3, that is, the β3a/β3b hairpin, strand β5 and helix α3 (details below).

All nts of the yU6 3′-overhang (G81-U90) contained in the present structure, except the first G81 residue, directly contact yPrp3$^{CTF}$ in one or both complexes, consistent with the important role of this part of U6 for stable Prp3 binding in yeast and human. In one yPrp3$^{CTF}$-yU4/U6$^{stem\ II+10nt}$ complex, the guanidinium group of R399 (helix α2) forms ionic interactions to the backbone phosphate of A82 and the side chain amino group of K355 (helix α1) binds to the A83 phosphate (*Figure 4A*). In the complex lacking these interactions, the corresponding backbone regions of yU6 and the stem II duplex are slightly pulled away from the protein, presumably by crystal packing interactions. The exocyclic N6 group of A83 approaches the side chain carboxyl of E362 (helix α1) and its base stacks on the side chain of F354 (helix α1; *Figure 4A*). The extracyclic amino group (N4) of C84 is hydrogen bonded to the backbone carbonyl of E407 (β3a/β3b hairpin) and the base is thereby held sandwiched between H409 (β3a/β3b hairpin) and P350 (helix α1; *Figure 4B*). The A83 and C84 backbone phosphates approach the side chain of K351 (helix α1; *Figure 4B*). These interactions splay out A83 and C84, stabilizing the first kink in the yU6 3′-overhang. The guanidinium group of R353 (helix α1) engages in hydrogen bonds to the Watson-Crick face of C85 (atoms O2 and N3) and to the Hoogsteen face of G86 (atoms O6 and N7; *Figure 4C*). Furthermore, N1 of G86 is contacted by the side chain carboxyl of D374 (β1-β2 loop), its base engages in cation-π stacking to the guanidium group of R371 (strand β2) and its 2′-hydroxyl group is hydrogen bonded to the backbone carbonyl of M442 (strand β5; *Figure 4C*). O4 of U87 is hydrogen bonded to the side chain amide of Q457 (helix α3) and its base stacks on F441 (strand β5; *Figure 4D*). The O2 atom of U88 hydrogen bonds with the backbone NH of W440 (strand β5), positioning the base laterally on the F441 side chain. The U87-U88 backbone region encircles the side chain of K357 (helix α1), which contacts the anionic phosphate oxygens of U87 and the O3 atom of U88 (*Figure 4D*). The interactions involving K357, W440, F441 and Q457 stabilize the second kink in the yU6 3′-overhang. The side chain amino group of K361 (helix α1) contacts the U89

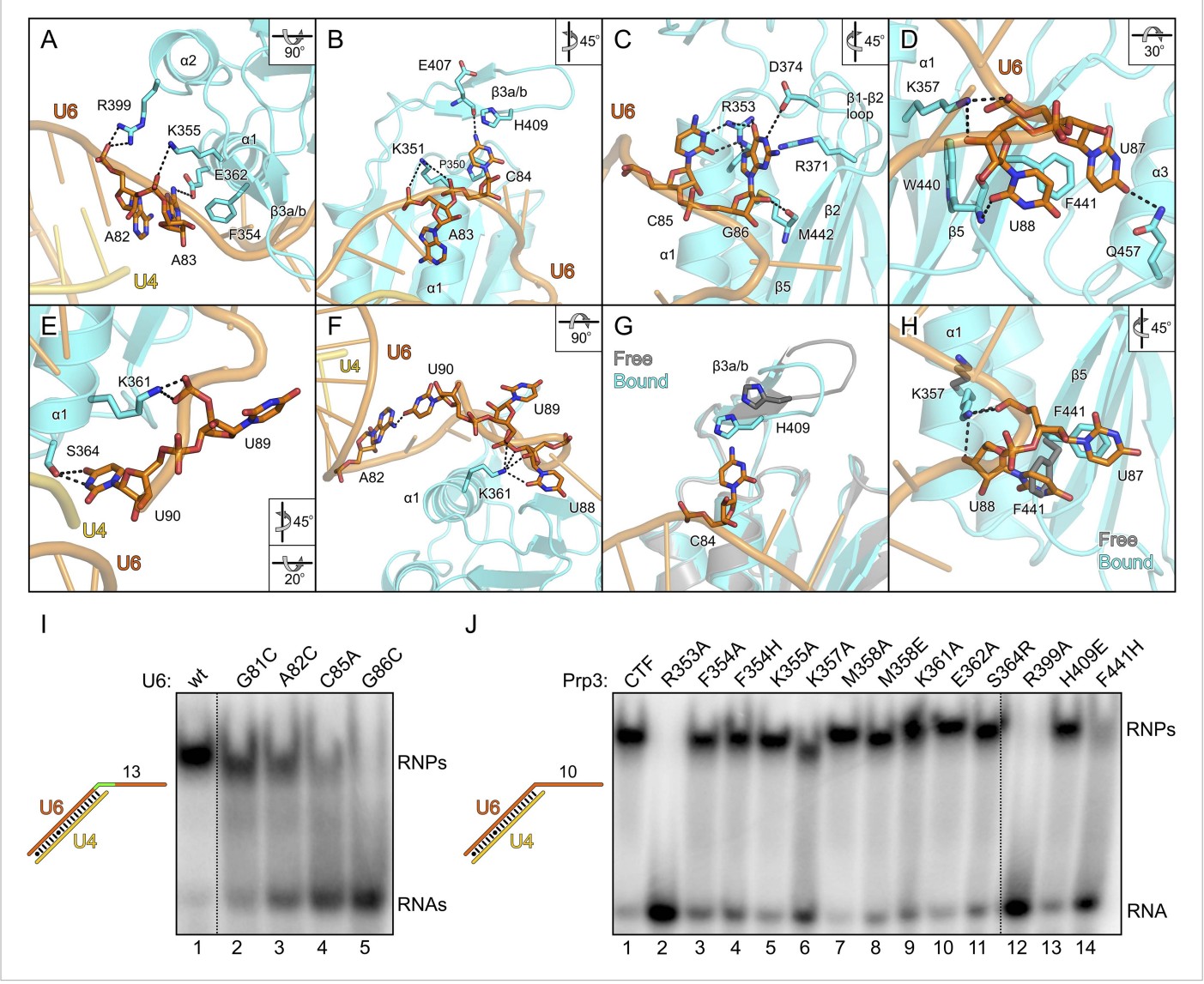

**Figure 4**. Details of the yPrp3-yU4/U6 di-snRNA interaction. (**A–H**) Close-up views of yPrp3–RNA contacts. The protein and the RNA are shown as semi-transparent cartoons (yPrp3$^{CTF}$—cyan; yU6—orange; yU4—gold) with interacting residues as sticks (colored by element; carbon or phosphorus—as the respective molecule; nitrogen—blue; oxygen—red, sulfur—yellow). Black dashed lines—hydrogen bonds or salt bridges. Panels (**G**) and (**H**) show overlays of unbound (protein—gray) and complex (protein—cyan; yU6 snRNA—orange) situations. Rotation symbols indicate the views relative to *Figure 3B*. (**I**) Binding of yPrp3$^{CTF}$ (20 µM) to wt and mutant versions (indicated above the gel) of U4/U6$^{stem\ II+13nt}$ (scheme on the left; mutated region in green). (**J**) Variants of yPrp3$^{CTF}$ (20 µM; indicated above the gel) binding to yU4/U6$^{stem\ II+10nt}$ (scheme on the left).

anionic phosphate oxygens in one complex (*Figure 4E*) and interacts with O2, O2′ and O3′ of U88 as well as the phosphate and O5′ of U89 in the other complex (*Figure 4F*). The last nt, U90, adopts two different conformations in the two complexes; in one case, its N3, O2 and O4 atoms are hydrogen bonded to the side chain hydroxyl of S364 (α3-β2 loop; *Figure 4E*), while in the other case it interacts with the N1 and N6 positions of nt A82 at the beginning of the yU6 3′-overhang (*Figure 4F*).

The overall structure of the DUF1115 domain is globally unchanged compared to the structure of isolated yPrp3$^{CTF}$ or yPrp3$^{DUF1115}$ (RMSD of 0.80–0.86 Å for 133 common Cα atoms) but there are local adjustments accompanying RNA binding, which may contribute to the binding specificity. Upon RNA binding, the β3a/β3b hairpin moves closer to the yPrp3$^{CTF}$ core domain to engage in direct interactions with the RNA and the H409 side chain in this element turns on top of the C84 base (*Figure 4G*). Within the yPrp3$^{CTF}$ core domain, the side chain of F441 is rotated ca. 90°C out of its

intra-molecular position in the isolated protein to stack on U87. Instead, the U88 base moves into the original F441 position, from where it contacts the W440 backbone (*Figure 4H*). Furthermore, the K357 side chain adopts an alternative conformation upon RNA binding to interact with the anionic phosphate oxygens of U87 and the O3′ atom of U88 (*Figure 4H*).

## Mutational probing of the protein–RNA interface

Although not directly contacted by the protein, the bases of G81 and A82 mediate a continuous π-stack from the terminal base pair of stem II to the yPrp3-bound portion of the yU6 3′-overhang. Consistent with this stacking being important for yPrp3 binding, replacement of these purines in yU4/U6$^{stem\ II+13nt}$ with a smaller pyrimidine (C) led to reduced affinity to yPrp3$^{CTF}$ (*Figure 4I*, lanes 2 and 3). Sequence-specific interactions are seen between R353 and the Watson-Crick edge of C85 as well as the Hoogsteen edge of G86 (*Figure 4C*), suggesting that the protein reads out parts of the RNA sequence. Consistently, mutation of C85 to A or G86 to C in yU4/U6$^{stem\ II+13nt}$ significantly reduced or essentially abrogated the interaction with yPrp3$^{CTF}$, respectively (*Figure 4I*, lanes 4 and 5). Correspondingly, a yPrp3$^{CTF}$ variant bearing a R353A exchange showed essentially no binding to yU4/U6$^{stem\ II+10nt}$ anymore (*Figure 4J*, lane 2; this and all other yPrp3$^{CTF}$ variants discussed below could be produced and purified like the wild type [wt] protein, suggesting that none of the tested mutations interfered with proper protein folding).

We also exchanged several additional yPrp3 residues that directly contact yU4/U6$^{stem\ II+10nt}$ in our structure and tested binding of the corresponding yPrp3$^{CTF}$ variants to this RNA (*Figure 4J*). R399A (lane 12) and F441H (lane 14) essentially abrogated the interaction with the RNA, while K357A (lane 6) strongly and F354A/H (lanes 3 and 4), K361A (lane 9) and S364R (lane 11) weakly destabilized the complex (indicated primarily by larger fractions of unbound yU4/U6$^{stem\ II+10nt}$ in the corresponding lanes). Single residue exchanges K355A (lane 5), M358A/E (lanes 7 and 8), E362A (lane 10) and H409E (lane 13) of yPrp3$^{CTF}$ showed no significant or only very mild reduction in RNA binding under the chosen conditions, indicating that individually the interactions involving these residues are not essential for stable RNA complex formation.

Nts G81, A82, C85 and G86, which upon mutation led to reduced yPrp3$^{CTF}$ binding, are highly evolutionarily conserved in U6 snRNAs (*Figure 1—figure supplement 2*). Likewise, R353, R399 and F441, where substitutions strongly affected RNA binding, are conserved in Prp3 orthologs (*Figure 1—figure supplement 1*). We conclude that the mode of Prp3-U4/U6 interaction seen in our structure is present in all eukaryotes.

## Evolutionary comparisons

The ferredoxin superfamily currently encompasses 59 subfamilies. Previous bioinformatics analyses indicated that within the superfamily the Prp3 DUF1115 domain is most homologous to acylphosphatase (AcyP) and BLUF domains (*Korneta et al., 2012*), which do not recognize nucleic acids. However, the sequence comparison did not reveal the evolutionary relationship between the Prp3 ssRNA-binding domain and other members of the ferredoxin superfamily. We therefore calculated a phylogenetic tree based on pairwise structural comparisons with representative ferredoxin fold structures (*Figure 3—figure supplement 2A*). The tree is in agreement with sequence analyses, showing that the Prp3 ssRNA-binding domain is most similar to AcyP and BLUF domains. For example, residues 335–467 of yPrp3$^{CTF}$ (*Figure 3—figure supplement 2B*) spatially align with cow AcyP (PDB ID 2ACY; *Figure 3—figure supplement 2C*) with a RMSD of 2.5 Å over 90 common Cα atoms (Z-score 10.1) and show a comparable similarity to the BLUF domain of the BlrB photoreceptor from *Rhodobacter sphaeroides* (PDB ID 2BYC; RMSD of 2.0 Å for 84 common Cα atoms; Z-score of 10.0; *Figure 3—figure supplement 2D*). On the other hand, the Prp3 domain is only distantly related to other ferredoxin fold proteins that bind RNAs, such as RRMs (*Figure 3—figure supplement 2F,G*) or ribosomal proteins S6, S10, and L10. The closest nucleic acid-binding relative of the Prp3 ssRNA-binding domain is the IS608 transposase domain, which recognizes DNA (*Figure 3—figure supplement 2H*).

## In vivo effects of Prp3 variants defective in U4/U6 di-snRNA binding

To test the importance of the observed yPrp3-yU4/U6 interactions for the function of yPrp3 in vivo, we used a haploid yeast strain, in which the chromosomal copy of the *PRP3* gene was deleted and the

essential protein was produced from a counter-selectable *URA3*-marked plasmid. We then shuffled plasmids carrying mutant *prp3* genes into this strain, selected against the *URA3* plasmid and monitored the effects of yPrp3 variants on cell viability, snRNP levels and pre-mRNA splicing.

yPrp3 proteins bearing mutations R304A, R322A or both (in the basic peptide preceding DUF1115) as well as variants bearing R399A or F441A exchanges (yU6 3′-overhang-binding DUF1115 domain) led to mild temperature sensitive (ts) growth (*Figure 5A*). A R399A/F441A double mutant strongly exacerbated this effect (*Figure 5A*). These observations are consistent with the mutated residues

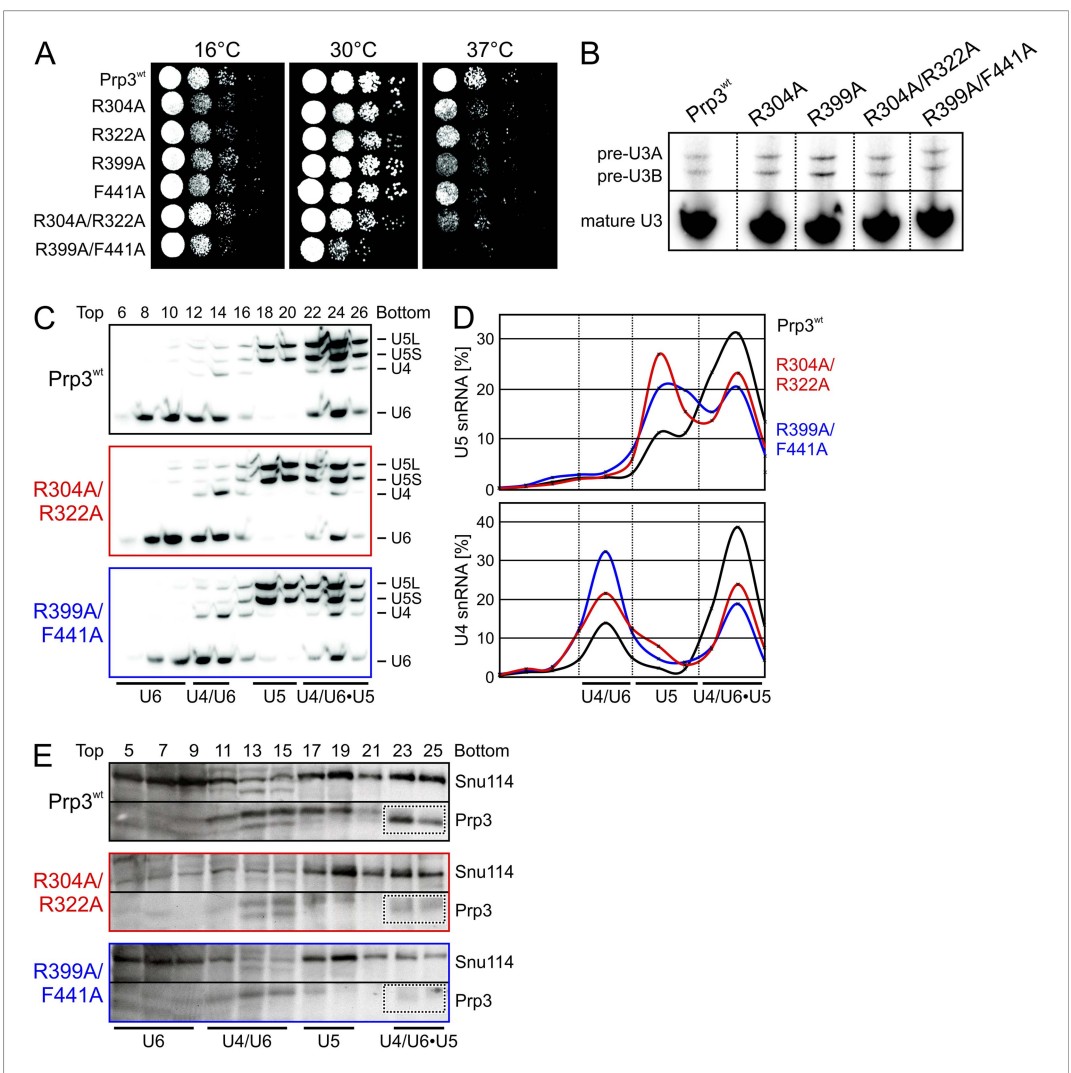

**Figure 5**. Consequences of yPrp3 variants in vivo. (**A**) Growth of yeast strains producing the indicated yPrp3 variants at various temperatures. Serial dilutions of liquid cultures were spotted on YPD agar plates and incubated at the indicated temperatures for 1–2 days. (**B**) In vivo splicing assays, monitoring levels of U3 pre-snoRNAs in yeast strains producing wt yPrp3 or the indicated yPrp3 variants. Cells were grown at 37°C for 5 hr before total RNA was extracted and U3 mature or pre-snoRNAs were detected by primer extension using a radiolabeled DNA oligonucleotide complementary to a region in U3 exon 2. (**C**, **D**)Northern blotting of cellular extracts of wt (top) or the indicated *prp3* mutant yeast strains (middle and bottom). Cellular extracts were separated on 10–30 % (v/v) glycerol gradients. Even-numbered gradient fractions (indicated above the blots) were probed with radiolabeled DNA oligomers complementary to snRNA regions. Positions of various snRNPs on the gradients are indicated below the blots. (**D**) Quantification of the data shown (**C**). (**E**) Western blots of the same gradients. Odd numbered gradient fractions (indicated above the blots) were probed with anti-ySnu114 (top) and anti-yPrp3 (bottom) antibodies. Positions of various snRNPs on the gradients are indicated below the blots. Dotted boxes—yPrp3 signals in the U4/U6•U5 tri-snRNP fractions.

contributing to functionally important yPrp3-yU4/U6 interactions, as seen in our binding assays and yPrp3$^{CTF}$ -yU4/U6$^{stem\ II+10nt}$ crystal structure. To trace the origin of the growth defects, we monitored the effects of the mutations on splicing in vivo at 37°C, using endogenous U3 (pre-)snoRNA as a reporter. Slight accumulation of unspliced U3 pre-snoRNAs was seen in strains producing the R304A, R399A, R304A/R322A or R399A/F441A variants of yPrp3 compared to the wt (*Figure 5B*), indicating that the observed growth defects likely root in inefficient splicing. Finally, monitoring snRNP levels by Northern blotting of cellular extracts spread out on a glycerol gradient showed that the levels of U4/U6 di-snRNP and of isolated U5 snRNP were increased at the expense of U4/U6•U5 tri-snRNP in strains producing Prp3 variants R304A/R322A and R399A/F441A (*Figure 5C,D*). At the same time, Western blotting revealed reduced levels of yPrp3 associated with U4/U6•U5 tri-snRNP in the mutant strains (*Figure 5E*). These results suggest that reduced tri-snRNP levels are the cause of the reduced splicing activity in the mutant strains and that efficient yPrp3 binding to U4/U6 di-snRNAs via its C-terminal region is important for U4/U6•U5 tri-snRNP stability.

## Discussion

### An evolutionarily conserved, composite ds/ssRNA-binding region in Prp3

Here, we showed that Prp3 orthologs contain a C-terminal U4/U6 di-snRNA binding region that encompasses a DUF1115 domain and a preceding peptide rich in basic residues. Our crystal structure and targeted mutational analyses demonstrate that the DUF1115 domain acts as a ssRNA-binding domain that specifically recognizes the first 10 nts of the U6 3′-overhang. While our structural analysis did not reveal how the preceding peptide is involved in U4/U6 binding, several pieces of evidence indicate that it binds along U4/U6 stem II: The peptide is important for full binding of Prp3 C-terminal portions to U4/U6 (*Figure 1C,D*) but it is not a structural element of the DUF1115 domain and thus does not act via stabilizing the RNA-binding fold of DUF1115. Single Arg-to-Ala exchanges in the peptide lead to reduced RNA binding (*Figure 1C,D*), indicating that these residues may directly contact the RNA. RNA constructs with shortened stem II or in which a conserved non-canonical base pair was converted to a Watson-Crick pair show reduced binding to Prp3$^{CTF}$ (*Figure 1E,F*), which cannot be explained by the observed DUF1115 domain contacts (exclusively to the U6 3′-overhang). Finally, Prp3$^{CTF}$, but not the DUF1115 domain, inhibits Brr2-mediated U4/U6 unwinding (*Figure 2A,B*), suggesting that the peptide stabilizes contacts between U4 and U6, presumably by binding the stem II duplex.

Residues in both the DUF1115 domain and the preceding basic peptide of Prp3 as well as nts in U4/U6, which we identified as being important for stable complex formation, are evolutionarily highly conserved, suggesting that the composite ds/ss U4/U6 di-snRNA-binding region of Prp3 is present in all eukaryotes. DUF1115 exhibits a ferredoxin-like core similar to AcyP/BLUF proteins but evolutionarily remote from other nucleic acid-binding domains in the ferredoxin superfamily. As we failed to detect a DUF1115-like domain by sequence comparisons in hundreds of other RNA-binding proteins recently identified in transcriptome-wide screens (*Baltz et al., 2012*; *Castello et al., 2012*), DUF1115 most likely represents a spliceosome-specific ssRNA-binding domain. Due to the pronounced reorientation that the U6 3′-overhang experiences on the surface of the Prp3 DUF1115 domain, the domain likely is an important architectural element in the U4/U6 di-snRNP and/or the U4/U6•U5 tri-snRNP.

### Prp3 binding helps explain defects associated with U6 snRNA variants

The mode of Prp3 binding to U4/U6 di-snRNAs revealed herein helps to rationalize a large body of mutational analyses on U6 snRNAs in yeast and human. Previously, the functional importance of various U6 snRNA regions was studied by site-directed mutagenesis. hU6 snRNA bearing a C62G mutation, which converts the C-U mismatch in U4/U6 stem II into a G-U wobble pair, supported only low levels of splicing (*Wolff and Bindereif, 1993*). Likewise, the corresponding C69G mutation in yU6 snRNA led to reduced splicing activity (*Ryan and Abelson, 2002*). We showed that a yU4/U6 stem II construct bearing the C69G mutation exhibits reduced binding to yPrp3 and according to our structural model this region of the U4/U6 di-snRNAs is recognized by the peptide preceding the Prp3 DUF1115 domain. Therefore, a defect in Prp3 binding may underlie the splicing defects of these U6 snRNA mutations in yeast and human.

Residues of the highly conserved UGA motif located in the terminus of U6 stem II and beginning of the 3′-overhang were shown to be important for U4/U6•U5 tri-snRNP stability and splicing.

Deletion of nts U74-G75-A76 in hU6 snRNA (corresponding to residues U80-G81-A82 in yU6) reduced the levels of U4/U6•U5 tri-snRNP and assembled spliceosomes to less than 10% of wt and abrogated splicing activity (*Wolff and Bindereif, 1992*). An A76C variant of hU6 snRNA showed reduced interaction with U4 snRNA, concomitant with reduced spliceosome assembly and low splicing activity (*Wolff and Bindereif, 1995*). Furthermore, yU6 snRNA bearing a U80G exchange almost completely abolished formation of U4/U6 di-snRNPs and U4/U6•U5 tri-snRNPs and was defective in splicing (*Fabrizio et al., 1989*; *Ryan and Abelson, 2002*; *Ryan et al., 2002*). The G81C point mutation in yU6 snRNA strongly blocked splicing in vitro (*Madhani et al., 1990*; *Ryan and Abelson, 2002*) and caused a substantial accumulation of free U6 snRNP deficient in U4/U6 di-snRNP assembly (*Ryan and Abelson, 2002*). The effects of some of these mutations were suggested to root in a stabilization of intra-molecular base pairing in U6 snRNA. However, our observation that G81C and A82C exchanges in yU6 snRNA led to weak binding of yPrp3 in vitro suggests that aberrant Prp3 binding may also contribute to these phenotypes. The continuous stacking of nts in the transition region between stem II and the U6 3′-overhang seen in our structure may properly orient these elements for stable Prp3 binding.

3′-truncated yU6 snRNAs containing residues 1–94, 1–91 or 1–88 allowed reconstitution of 35–40 % of wt splicing activity, while only 20% and 4% of the wt splicing activity were regained with yU6 snRNA molecules further shortened to residues 1–86 or 1–85, respectively, and yU6 snRNAs containing only residues 1–81 or 1–80 fully blocked splicing (*Madhani et al., 1990*; *Ryan et al., 2002*). The phenotypes associated with the deletion of distal yU6 3′-overhang residues likely originate from the removal of binding sites for the Prp24 assembly chaperone and the LSm protein complex. However, the exacerbated effects when yU6 snRNA was shortened to 86 nts or less correlate very well with our finding of reduced yPrp3 binding to yU4/U6 constructs bearing sequentially shortened U6 3′-overhangs or the C85A and G86C point mutations. Reduced Prp3 binding may also contribute to the reduced hU4/U6 di-snRNP levels seen previously in the presence of hU6 snRNA bearing a C79U exchange in the 3′-overhang (*Wolff and Bindereif, 1995*), as we observed weak yPrp3 interaction with a yU4/U6 bearing a variant residue at the analogous position (C85A).

## Mechanism of Prp3-dependent U4/U6•U5 tri-snRNP stability

The binding of the U4/U6-specific proteins Snu13, Prp31 and the Prp3-Prp4(-CypH) complex to U4/U6 di-snRNAs is highly interdependent. Snu13 binds a K-turn motif in the U4 5′SL (*Nottrott et al., 1999*; *Vidovic et al., 2000*), serving as an assembly initiating protein for the subsequent incorporation of the other components (*Nottrott et al., 2002*). An explanation for the ordered binding of Snu13 and Prp31 was provided by structural studies that showed how hSnu13 and the hU4 5′SL provide a composite binding platform for the hPrp31 Nop domain (*Liu et al., 2007*, *2011*), while the mechanism by which hSnu13 leads to enhanced interaction of hPrp3 with U4/U6 di-snRNAs (*Nottrott et al., 2002*) remained enigmatic. Our results indicate that Prp3 contacts RNA elements along stem II, that is, in the vicinity of the U4/U6 3-way junction. Snu13 binds one branch of the 3-way junction (the U4 5′SL), which could exert a stabilizing effect on the 3-way junction and stem II and lead to improved binding of Prp3. This effect could be further enhanced by Prp31, whose U4/U6 contacts extend into the lower part of stem II (*Schultz et al., 2006*). It is also conceivable that the Prp3 stem II-binding peptide directly contacts Snu13 and/or Prp31 on the RNAs and that the latter two proteins even modulate how the peptide interacts with stem II.

Our in vivo analyses of the effects of yPrp3 mutations that interfere with stable yU4/U6 binding in vitro, show that the yPrp3-yU4/U6 interactions we observe in our complex structure are important for U4/U6•U5 tri-snRNP stability and, likely as a consequence, for pre-mRNA splicing. The role of Prp3 in mediating tri-snRNP stability can be understood from our present data, showing that the C-terminal portions of Prp3 maintain interactions with the U4/U6 di-snRNAs, and previous yeast 2-hybrid (Y2H) analyses, showing that Prp3 contacts the U5 snRNP proteins Prp6 and Snu66 (*Liu et al., 2006*). By combining these functions, Prp3 apparently acts as a bridge, linking the U4/U6 di-snRNP and the U5 snRNP via Prp3-RNA interactions on the U4/U6 side and Prp3-protein interactions on the U5 side (*Figure 6A*).

## Possible role of Prp3 in U4/U6 reassembly after splicing

After their release from the spliceosome, U4 and U6 snRNAs have to be re-decorated with proteins and reassembled into a di-snRNP before they can participate in further rounds of splicing. This recycling requires the Prp24 protein in yeast and the related SART3 protein in human. Recently, a crystal structure

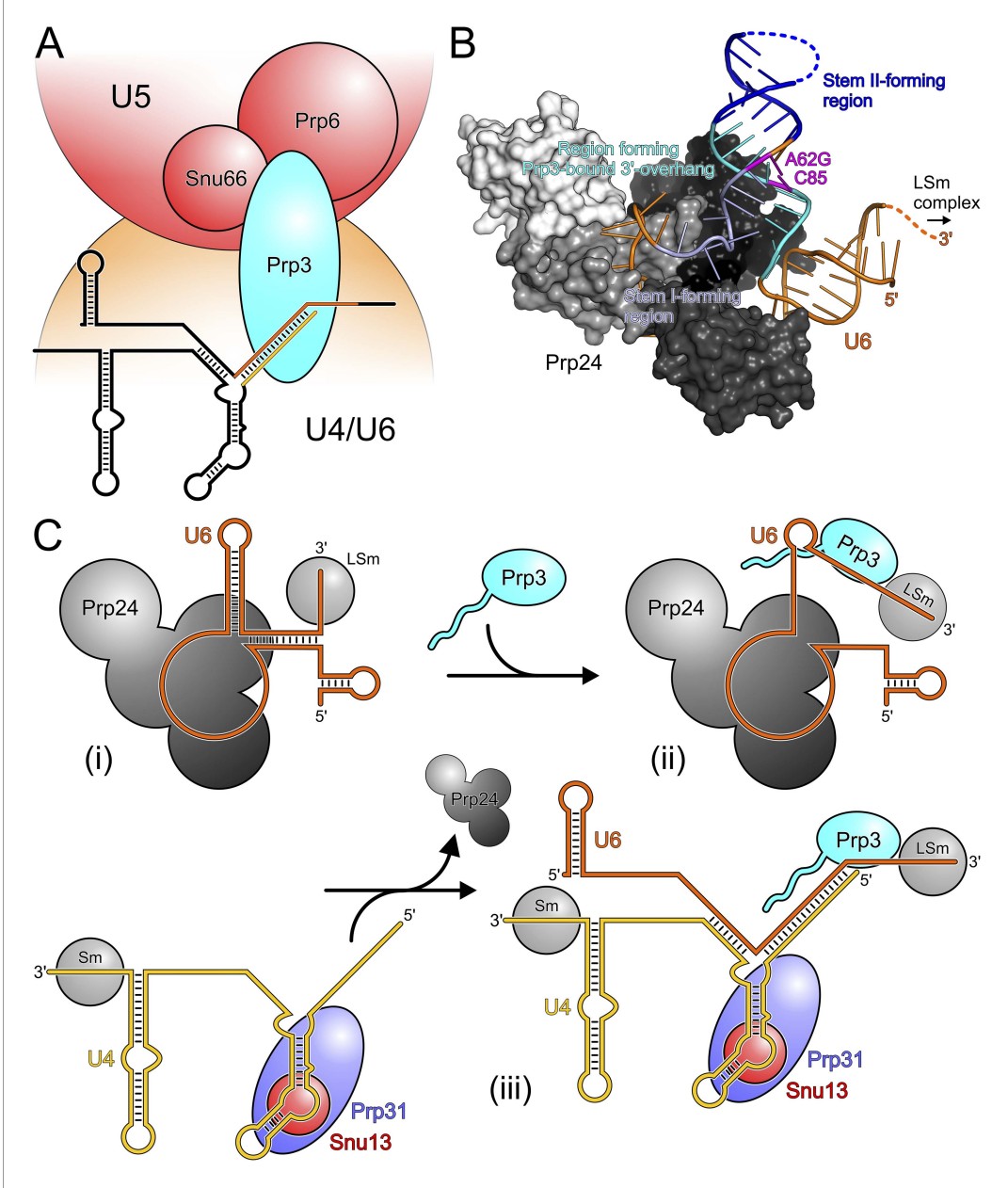

**Figure 6**. Models for splicing-associated functions of Prp3. (**A**) Function of Prp3 as a bridge in the U4/U6•U5 tri snRNP. U4/U6 di-snRNP—orange; U5 snRNP—red; snRNAs—black sticks; U4/U6 stem II and U6 3'-overhang—gold and orange sticks. Prp3 uses its C-terminal region to bind U4/U6 stem II and the U6 3'-overhang on the U4/U6 di-snRNP side (this work) and interacts with proteins Prp6 or Snu66 on the U5 snRNP side (*Liu et al., 2006*). (**B**) Structure of yPrp24 in complex with yU6 snRNA (*Montemayor et al., 2014*). yPrp24 RRM 1–4 domains—white, light gray, dark gray and black, respectively. Regions of yU6 forming stem I, stem II and the 3'-overhang in yU4/U6 are shown in different blue colors. Cold-sensitive A62G mutation and its paired nucleotide, C85—magenta. (**C**) Model for the function of Prp3 during U4/U6 di-snRNP reassembly after splicing. (i) Upon release from the spliceosome, U6 snRNA is bound by Prp24 and LSm proteins. (ii) Recruitment of Prp3 and association with its cognate U6 regions, which are partially exposed in the Prp24-U6 complex. Emerging Prp3-U6 interactions initiate detachment of U6 snRNA from Prp24. (iii) Incorporation of pre-assembled U4 snRNP may complete Prp24 displacement and assembly of U4/U6 di-snRNP. U4 incorporation may be aided by Prp3 stabilizing the emerging stem II and by the cooperative binding of Snu13, Prp31 and Prp3 to U4/U6 di-snRNAs.

of near full-length Prp24 in complex with a large portion of U6 snRNA was elucidated (*Montemayor et al., 2014*). While not contained in the Prp24-U6 structure, the distal 3′-end of U6 snRNA is available in that complex for binding the LSm proteins (*Figure 6B*), consistent with the observation of a stable Prp24-LSm-U6 snRNA complex in yeast (*Vidal et al., 1999*). In contrast, U6 regions that form stem I, stem II and the U6 3′-overhang in the assembled U4/U6 di-snRNP are sequestered in an internal U6 stem loop (ISL) and by Prp24-U6 interactions (*Figure 6B*). As Prp24 does not harbor an NTP-dependent RNA helicase activity, the question arises how the U6 ISL can be unwound and U6 handed over from Prp24 to U4 snRNP. Notably, regions of U6 snRNA that form stem II and the U6 3′-overhang in the U4/U6 di-snRNP (i.e., Prp3-binding elements) are largely exposed on the surface of the Prp24-U6 snRNA complex (*Figure 6B*). Furthermore, the Prp24-U6 structure represents an artificially stabilized situation, obtained after introducing a cold-sensitive A62G mutation in U6, which impedes unwinding of the U6 ISL (*Fortner et al., 1994*) by sequestering C85 in a non-natural Watson–Crick base pair (magenta in *Figure 6B*). We have shown here that C85 is a crucial contact residue of Prp3, which also latches onto the surrounding nts in the context of U4/U6 di-snRNP via its DUF1115 domain. Thus in the wt situation, the U6 ISL on Prp24 can most likely "breathe" in the region surrounding C85, whereby this region could become available for Prp3 binding. Consistent with this idea, chemical probing studies showed that these Prp3-binding nts are only weakly protected in the Prp24-LSm-U6 snRNA particle (*Karaduman et al., 2006*). We therefore suggest that binding of Prp3 to its cognate U6 3′-overhang region traps and subsequently extends and stabilizes spontaneous local unwinding of the internal stem loop (ISL) on the Prp24-LSm-U6 snRNA complex (*Figure 6C*).

We envision that Prp24 displacement is completed by entry of a pre-assembled U4 snRNP (*Figure 6C*). Prp3 could also support this step by stabilizing the emerging U4/U6 stem II via its stem II-binding peptide. Furthermore, the cooperativity we detect in binding of Snu13, Prp31 and Prp3 to U4/U6 di-snRNA may help to complete assembly of the U4/U6 di-snRNP. Taken together, our results support an important role of Prp3 in the handover of U6 snRNA from Prp24 to U4 snRNP during U4/U6 di-snRNP reassembly after splicing.

## Materials and methods

### Protein production and purification

DNA encoding full-length hPrp3 or the hPrp3$^C$ fragment (residues 195–683) with C-terminal His$_6$-tags were PCR-amplified from pGADT7-*hPRP3* (*Liu et al., 2006*) and subcloned into vector pMAL-c2t, in which the Factor Xa cleavage site following the N-terminal MBP tag of pMAL-c2x (New England Biolabs, Ipswich, MA) was replaced by a TEV cleavage site. The inserts of these and all other plasmids were verified by sequencing. A DNA fragment encoding hPrp3$^N$ (residues 1–442) was cloned into pETM-41 (EMBL, Heidelberg, Germany) for production of the protein bearing an N-terminal His$_6$-MBP fusion. Plasmids were transformed into *Escherichia coli* BL21 (DE3)-RIL cells (Stratagene, La Jolla, CA) and expressed at 16–18°C using the auto-induction method (*Studier, 2005*). Target proteins were double affinity purified using Ni-NTA beads and amylose resin followed by gel filtration using a Superdex 200 26/60 column (GE Healthcare, Munich, Germany) in 20 mM Tris–HCl, pH 7.5, 200 mM NaCl, 2 mM DTT. RNase A (Qiagen, Hilden, Germany) was included in the initial purification steps for digestion of host RNAs.

DNA fragments encoding hPrp3$^{CTF}$ (residues 484–683) or yPrp31$^{1–462}$ were introduced via BamHI and XhoI restriction sites into pGEX-6P (GE Healthcare) for production as N-terminal, PreScission-cleavable GST fusion proteins. GST-hPrp3$^{CTF}$ and GST-yPrp31$^{1–462}$ were produced at 18°C in *E. coli* BL21(DE3)-RIL cells using the auto-induction method and purified using glutathione-Sepharose beads (GE Healthcare). The fusion proteins were eluted from beads in buffer containing 10 mM reduced L-glutathione and further purified via a Superdex 75 16/60 gel filtration column (GE Healthcare) in 20 mM Tris–HCl, pH 7.5, 200 mM NaCl, 2 mM DTT.

DNA constructs encoding yPrp3$^{CTF}$ (residues 296–469), yPrp3$^{DUF1115}$ (residues 325–469) or ySnu13 were amplified from *PRP3* or *SNU13* synthetic genes (GENEART AG, Regensburg, Germany), respectively, and subcloned into pETM-11 (EMBL, Heidelberg). Mutant versions of pETM-11-*PRP3$^{CTF}$* were generated by site-directed mutagenesis using the QuikChange II kit (Agilent Technologies, Santa Clara, CA). The plasmids were transformed into *E. coli* BL21 (DE3)-RIL cells and expressed at 18–25°C for two days using the auto-induction method. N-terminally His$_6$-tagged proteins were affinity purified using 1 ml or 5 ml HisTrap FF columns (GE Healthcare). 1 M LiCl buffer was included in

the wash step to remove unspecifically bound nucleic acids. His$_6$-tagged proteins used in some EMSA and unwinding assays were directly applied to a Superdex 75 16/60 column in 10 mM Tris–HCl, pH 7.0, 150 mM NaCl, 2 mM DTT. For EMSA titrations, ITC and crystallization, TEV protease was used to remove the N-terminal His$_6$-tag before the size-exclusion chromatography step. Production of selenomethionine (SeMet)-substituted His$_6$-yPrp3$^{CTF}$ in BL21 (DE3)-RIL cells was carried out in M9 minimal medium supplemented with trace metals and vitamins (*van den Ent and Lowe, 2000*). At an OD$_{595}$ of 0.6, 50 mg/l of SeMet (Fisher Scientific, Hampton, NH), 100 mg/l of lysine, threonine and phenylalanine and 50 mg/l of leucine, isoleucine and valine were added. After 15 min, the temperature was reduced to 20°C and 0.32 mM IPTG was added for induction overnight. SeMet-substituted protein was purified in the same way as the native protein.

Full-length yBrr2 bearing an N-terminal His$_6$-tag was produced using a recombinant baculovirus in insect cells as described for hBrr2 (*Santos et al., 2012*). Briefly, a 800 ml infected High Five™ cell pellet was resuspended in 50 mM HEPES, pH 8.0, 600 mM NaCl, 2 mM β-mercaptoethanol, 0.05% NP-40, 20% glycerol, 10 mM imidazole, supplemented with protease inhibitors (Roche, Penzberg, Germany) and lyzed by sonication using a Sonopuls Ultrasonic Homogenizer HD 3100 (Bandelin). His$_6$-yBrr2 was captured from the cleared lysate on a 5 ml HisTrap FF column (GE Healthcare) and eluted with a linear gradient from 10 to 250 mM imidazole. The eluted protein was diluted to a final concentration of 80 mM sodium chloride and loaded on a Mono Q 10/100 GL column (GE Healthcare) equilibrated with 50 mM Tris–HCl, pH 8.0, 50 mM NaCl, 5% glycerol, 2 mM β-mercaptoethanol. His$_6$-yBrr2 was eluted with a linear 50 to 600 mM sodium chloride gradient and further purified by gel filtration on a Superdex 200 26/60 column (GE Healthcare) in 40 mM Tris–HCl, pH 8.0, 200 mM NaCl, 20% glycerol, 2 mM DTT.

## RNA production and purification

Full-length yU4 and yU6 snRNAs were synthesized by in vitro transcription using T7 RNA polymerase and PCR-generated templates. The transcripts were purified using the RNeasy Midi Kit (Qiagen). RNA duplexes were prepared by combining 30 nM of [$^{32}$P]-5′-end labeled U4 snRNA with a fivefold molar excess of unlabeled U6 snRNA in annealing buffer (40 mM Tris–HCl, pH 7.5, 100 mM NaCl). The solution was heated to 80°C for 2 min and cooled to 25°C over 90 min. 12.5 mM MgCl$_2$ were added to the solution at 70°C. The annealed duplex was separated from ss U4 and U6 snRNAs by 6% native PAGE. The duplex was eluted from the gel, phenol-chloroform extracted, ethanol precipitated and resuspended in annealing buffer. yU4/U6$^{fused}$ was prepared by in vitro transcription and purified via a Mono Q column, followed by gel electrophoresis on an 8% denaturing (7 M urea) polyacrylamide gel, eluted and precipitated by isopropanol. All other RNAs were chemically synthesized (Dharmacon, Lafayette, CO). Complementary oligonucleotides were annealed before use by resuspension in H$_2$O, mixing, heating to 95°C for 2 min, slow cooling to room temperature followed by incubation on ice.

## Electrophoretic mobility shift assays

Typically, [$^{32}$P]-5′-end labeled RNAs were mixed with recombinant proteins in 20 mM Tris–HCl, pH 7.0, 150 mM NaCl, 2 mM DTT, 0.5 μg/μl tRNA. For EMSAs involving hPrp3$^{FL}$, hPrp3$^N$ and hPrp3$^C$, samples were incubated in 10 mM Tris–HCl, pH 7.5, 200 mM NaCl, 2 mM DTT, 0.33 μg/μl tRNA and 0.2 μg/μl heparin. For hPrp3$^{CTF}$ the same buffer with 67 ng/μl heparin was used. RNP complexes were allowed to form for 30 min at 4°C and then separated on a 5–6 % (60:1) polyacrylamide gel.

For EMSA titrations of yPrp3$^{CTF}$ variants and yPrp3$^{DUF1115}$, proteins lacking the N-terminal His$_6$-tag were employed. EMSA experiments were performed as described above (pre-incubation in 20 mM HEPES, pH 6.8, 150 mM NaCl, 0.33 μg/μl tRNA) with the indicated concentrations of proteins. Radioactive bands were visualized by autoradiography and quantified with the Image Quant 5.2 software (GE Healthcare). Apparent K$_d$ values were obtained by fitting the resulting data points to a single exponential Hill equation (fraction bound = A[protein]$^n$/([protein]$^n$ + K$_d$$^n$); A, fit maximum of RNA bound; n, Hill coefficient) (*Ryder et al., 2008*) using GraphPad Prism (GraphPad Software, Inc., La Jolla, CA).

For monitoring binding cooperativity of yPrp3$^{CTF}$ and yPrp3$^{CTF}$ with Snu13 and Prp31$^{1–462}$, [$^{32}$P]-5′-end labeled RNA oligonucleotides were mixed with recombinant proteins in 20 mM HEPES-NaOH, pH 7.5, 200 mM NaCl, 3 mM DTT, 0.33 μg/μl tRNA, 67 ng/μl acetylated BSA, 13 ng/μl heparin at 4°C for 30 min and separated on a 4% (30:1) polyacrylamide gel. Radioactive bands were visualized by autoradiography using a phosphoimager (Molecular Dynamics, Sunnyvale, CA).

## Isothermal titration calorimetry

Proteins without affinity tags were used for ITC experiments. Proteins and RNA were buffer exchanged to 20 mM HEPES, pH 6.8, 150 mM NaCl by dialysis and their concentrations were determined via the absorbances at 280 and 260 nm, respectively. yPrp3$^{CTF}$ variants (50 µM) or yPrp3$^{DUF1115}$ (200 µM) were used as titrants, yU4/U6$^{stem\ II+13nt}$ (fused by a GAGA tetraloop; 300 µl at 5 µM or 25 µM) as the analyte. Measurements were conducted at 20°C on a MicroCal$^{TM}$ iTC200 (GE Healthcare), with 16 injections of 2.5 µl each with 180 s intervals between injections. Titrant heats of dilution were subtracted and data were fit using MicroCal Origin 7 (GE Healthcare).

## U4/U6 di-snRNA unwinding assays

1.75 nM yU4/U6 di-snRNAs without or with 20 µM of yPrp3 variants were pre-incubated with RNA at 30°C for 3 min in 40 mM Tris–HCl, pH 7.5, 50 mM NaCl, 8% glycerol, 0.5 mM MgCl$_2$, 100 ng/µl acetylated BSA, 1 U/µl RNasin, 1.5 mM DTT before the addition of 50 nM yBrr2. The reactions were initiated by adding 1 mM ATP/MgCl$_2$ and further incubated at 30°C for 0–90 min. Aliquots were taken at the indicated time points and reactions were stopped with one volume of 40 mM Tris–HCl, pH 7.4, 50 mM NaCl, 25 mM EDTA, 1% SDS, 10% glycerol, 0.05% xylene cyanol, 0.05% bromophenol blue. Samples were loaded on a 6% native PAGE and run at 10 W for 2 hr. RNA bands were visualized by autoradiography and quantified with the Image Quant 5.2 software. Data were fit to a single exponential equation (fraction unwound = A{1-exp(-$k_u$ t)}); A, amplitude of the reaction; $k_u$, apparent first-order rate constant for unwinding; t, time using GraphPad Prism (GraphPad Software, Inc.).

## Plasmid shuffling, cell viability, and in vivo splicing assays

The wt yPRP3 gene was PCR-amplified from plasmid EWB2235, cloned into vector pRS314 and used to introduce the desired mutations by the QuikChange site-directed mutagenesis strategy (Stratagene). Wt and mutant plasmids were transformed into yeast strain EWY2845 (prp3::LEU2; his3Δ200; leu2-3112; lys2-810; trp1-1; ura3-52 [PRP3/YCp50]; kindly provided by John L. Woolford, Jr., Carnegie Mellon University, Pittsburgh, USA) that harbors the PRP3 gene on a counter-selectable URA3 plasmid. Transformants were selected in a medium lacking tryptophan, followed by three times streaking on 5-FOA plates at 25°C to counter-select the URA3 plasmid. Growth phenotypes of the yeast cells were assessed by spotting about 5 × 10$^5$ cells and tenfold serial dilutions on YPD agar plates and incubating at 16°C, 30°C or 37°C for 1–2 days. To analyze the effect of prp3 mutations on splicing in vivo, the yeast cells producing wt or variant forms of yPrp3 were used to inoculate YPD medium at an OD$_{600}$ of 0.05 and the cultures were further incubated at 37°C for 5 hr. For monitoring the levels of unspliced and spliced U3 (pre-)snoRNAs, 8 µg of total RNA from each sample were used for primer extension as described (Mozaffari-Jovin et al., 2013).

## Analysis of snRNP levels

Yeast cells expressing wt or ts variants of yPrp3 were grown in YPD medium at 30°C. Whole cell extract prepared from each strain was incubated at 37°C (the non-permissive temperature for the Prp3 ts variants) for 30 min, diluted with equal volume of G100 buffer (20 mM HEPES-KOH, pH 7.0, 100 mM KCl, 0.2 mM EDTA) and fractionated by ultracentrifugation on a 12 ml 10–30 % (v/v) glycerol gradient in G100 buffer. Subsequent to ultracentrifugation at 37,000 rpm for 15 hr in a Sorvall TST41.14 rotor, the distribution of spliceosomal U4, U5 and U6 snRNPs across the gradient fractions was monitored by Northern blotting and quantified as described previously (Mozaffari-Jovin et al., 2013). The association of yPrp3 with tri-snRNP was analyzed by Western blotting of gradient fractions and immunostaining using antibodies against yPrp3 and ySnu114 and the Amersham ECL detection kit (GE Healthcare).

## Crystallographic procedures

Crystallization was performed using the sitting-drop vapor-diffusion method at 18°C. Protein yPrp3$^{CTF}$ was concentrated to 15 mg/ml in 10 mM Tris–HCl, pH 7.5, 200 mM NaCl, 1 mM DTT. The best crystals were obtained by mixing 1 µl of protein solution with 0.2 µl of 0.1 M yittrium (III) chloride hexahydrate or praseodymium (III) acetate hydrate and 0.8 µl of reservoir solution (0.1 M succinic acid, pH 7.0, 16.4% PEG 3350). Protein yPrp3$^{325–469}$ was concentrated to 15 mg/ml in 10 mM HEPES, pH 6.8, 100 mM NaCl, 2 mM DTT. The best crystals were obtained using a 1:1 mixture of protein solution and reservoir buffer consisting of 0.1 M HEPES, pH 7.5, 10% PEG 8000. Prior to data collection, both types of crystals were cryoprotected in reservoir solution supplemented with 27% (v/v) ethylene glycol and flash-cooled in liquid nitrogen.

yU4/U6$^{stem\ II+10nt}$ was slowly added to the SeMet-substituted yPrp3$^{CTF}$ in a 1:1 molar ratio and incubated at 4°C for 15 min. The mixture was then chromatographed on a Superdex 75 16/60 column in 10 mM Na HEPES, pH 6.8, 1 mM DTT. Peak fractions containing the yPrp3$^{CTF}$-yU4/U6$^{stem\ II+10nt}$ complex were pooled and concentrated to 7.5 mg/ml. Crystals were grown by mixing 1 µl of complex solution with 0.2 µl of 0.5 M sodium fluoride and 0.8 µl of reservoir solution (0.22 M DL-malic acid, pH 6.8, 16.5% PEG 3350). For diffraction data collection, crystals were transferred to a 1:1 mixture of Paratone-N and paraffin oil as a cryoprotectant and flash-cooled in liquid nitrogen.

All X-ray diffraction data were collected at 100 K on beamline BL14.2 at the BESSY II storage ring (Berlin, Germany). Data were processed with the HKL package (*Minor et al., 2006*) or XDS (*Kabsch, 2010*). The structure of yPrp3$^{CTF}$ was solved by a praseodymium (III) multiple anomalous dispersion experiment as described (*Puehringer et al., 2012*) and refined against the higher resolution data from an yttrium (III)-derivatized crystal. The structures of yPrp3$^{DUF1115}$ and of a yPrp3$^{CTF}$-yU4/U6$^{stem\ II+10nt}$ complex were solved by molecular replacement using structure coordinates of yPrp3$^{CTF}$ as search models with the programs MOLREP (*Vagin and Teplyakov, 2010*) and PHASER (*McCoy, 2007*), respectively. Models were manually completed and adjusted with COOT (*Emsley et al., 2010*) and the structures were refined using REFMAC5 (*Murshudov et al., 2011*) and phenix.refine (*Afonine et al., 2012*). To preserve A-form geometry in the stem II portion of the yPrp3$^{CTF}$-yU4/U6$^{stem\ II+10nt}$ structure, an A-form duplex was used as a reference model and the A-form geometry was restrained during the refinement. Electrostatic surface potentials were calculated with APBS (*Unni et al., 2011*). All structure figures were created using PyMOL (http://www.pymol.org/).

## Phylogenetic analyses

For phylogenetic analyses, a representative set of protein structures containing ferredoxin-like folds was assembled using the SCOP database (*Murzin et al., 1995*). Pairwise structural similarity Z-scores were calculated for all structures and the yPrp3 ssRNA-binding domain using DaliLite (*Holm and Park, 2000*). The Z-score reciprocals were used to build a UPGMA tree with the neighbor tool of the PHYLIP package (*Felsenstein, 1989*). The tree was visualized with Evolview (*Zhang et al., 2012*) and annotated manually.

## Data deposition

Coordinates and diffraction data have been deposited with the Protein Data Bank (www.pdb.org) under accession codes 4YHU (yPrp3$^{CTF}$), 4YHV (yPrp3$^{DUF1115}$) and 4YHW (yPrp3$^{CTF}$-yU4/U6$^{stem\ II+10nt}$) and will be released upon publication.

## Acknowledgements

We thank Nicole Holton, Freie Universität Berlin, for help with ITC measurements and Grzegorz Chojnowski, International Institute of Molecular and Cell Biology, for help with RNA structure comparisons. We thank Jelena Jakovljevic and John L Woolford, Jr, Carnegie Mellon University, Pittsburgh, USA, for yeast strain EWY2845 and plasmid EWB2235. We acknowledge access to beamline BL14.2 of the BESSY II storage ring (Berlin, Germany) via the Joint Berlin MX-Laboratory sponsored by the Helmholtz Zentrum Berlin für Materialien und Energie, the Freie Universität Berlin, the Humboldt-Universität zu Berlin, the Max-Delbrück Centrum, and the Leibniz-Institut für Molekulare Pharmakologie. MT was supported by a fellowship from the Boehringer Ingelheim Fonds. This work was funded by the Deutsche Forschungsgemeinschaft (SFB 740 to MCW; SFB 860 to RL), the Bundesministerium für Bildung und Forschung (grant 05K10KEC to MCW), the European Research Council (grant RNA + P = 123D to JMB) and the Foundation for Polish Science (to JMB).

## Additional information

### Funding

| Funder | Grant reference | Author |
| --- | --- | --- |
| Deutsche Forschungsgemeinschaft (DFG) | SFB 740 | Markus C Wahl |
| Bundesministerium für Bildung und Forschung (Federal Ministry of Education and Research) | 05K10KEC | Markus C Wahl |

| Funder | Grant reference | Author |
|---|---|---|
| European Research Council (ERC) | RNA+P=123D | Janusz M Bujnicki |
| Foundation For Polish Science (Fundacja na rzecz Nauki Polskiej) | "Ideas for Poland" fellowship | Janusz M Bujnicki |
| Boehringer Ingelheim Fonds (BIF) | None | Matthias Theuser |
| Deutsche Forschungsgemeinschaft (DFG) | SFB 860 | Reinhard Lührmann |

The funders had no role in study design, data collection and interpretation, or the decision to submit the work for publication.

## Author contributions

SL, Acquisition of data, Analysis and interpretation of data, Drafting or revising the article; SM-J, JW, KFS, MT, SD-H, PF, JMB, Acquisition of data, Analysis and interpretation of data; RL, MCW, Conception and design, Analysis and interpretation of data, Drafting or revising the article

# Additional files

### Major datasets

The following datasets were generated:

| Author(s) | Year | Dataset title | Dataset ID and/or URL | Database, license, and accessibility information |
|---|---|---|---|---|
| Liu S, Wahl MC | 2015 | yPrp3$^{CTF}$ | http://www.rcsb.org/pdb/search/structidSearch.do?structureId=4YHU | Publicly available at the RCSB Protein Data Bank (Accession no: 4YHU). |
| Liu S, Wahl MC | 2015 | yPrp3$^{DUF1115}$ | http://www.rcsb.org/pdb/search/structidSearch.do?structureId=4YHV | Publicly available at the RCSB Protein Data Bank (Accession no: 4YHV). |
| Liu S, Wahl MC | 2015 | yPrp3$^{CTF}$-yU4/U6$^{stem\ II+10nt}$ | http://www.rcsb.org/pdb/search/structidSearch.do?structureId=4YHW | Publicly available at the RCSB Protein Data Bank (Accession no: 4YHW). |

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
