## [Decision Letter]

Thank you for sending your work entitled “A composite double-/single-stranded RNA-binding region in protein Prp3 supports tri-snRNP stability and splicing” for consideration at *eLife*. Your article has been evaluated by James Manley (Senior editor) and three reviewers, one of whom is a member of our Board of Reviewing Editors. The Reviewing editor and the other reviewers discussed their comments before we reached this decision, and the Reviewing editor has assembled the following comments to help you prepare a revised submission.

All reviewers thought that your findings regarding the recognition of the 3' single-stranded region of U6 snRNA by Prp3 are a strong point. This represents a significant advance in our understanding of the mechanistic details of the spliceosome machinery, in particular in describing its complex structural changes, including the recycling phase.

However, as you will see in the reviews below, there were also major differences of opinion regarding the work, including serious criticism. The hydroxyl radical footprinting results (Figure 4) were considered not very convincing, and should be replaced with another experiment or removed with a corresponding softening of the relevant conclusion. Regarding the U4-U6 stem-Prp3 interaction, further experimental details should be added, along the lines suggested by Reviewer 3, to strengthen this point. There was also some criticism on the presentation of the data, and both the Introduction, the first part of the Results, and some of the Discussion sections were considered difficult to read, fraught with too many details. Some of this is due to confusing switches between human and yeast nomenclature, and would require considerable rewriting and streamlining.

Reviewer #1:

This manuscript presents crystallographic analysis of structures of Prp3 a U4/U6 di-snRNP. Structures of both Prp3 alone and bound to U4/U6 are presented. While some useful new information is obtained, e.g. what appears to be a novel RNA biding motif in Prp3, I have some concerns about the impact of the work. These include the fact that interaction between Prp3 and stem II of U4/U6 is not well resolved. Additionally, the relevance of these structures in the absence of Snu13 and Prp31 is questionable given that the three proteins bind U4/U6 cooperatively. It seems quite possible that the presence of Snu13 and or Prp31 could alter the functionally relevant conformation of Prp3.

Reviewer #2:

U4 and U6 small nuclear RNAs (snRNAs) undergo several conformational transitions during the spliceosome cycle, from the U4-U6 base pairing interaction, assembly into the U4/U6-U5 tri-snRNP, U4/U6-to-U6/U2 transition during splicing catalysis and U4 release, to post-spliceosomal recycling and re-assembly of the U4/U6-paired structure. The authors have made important contributions here in their previous work, revealing the protein components, the RNP structures and compositional/structural changes of all stages.

Here they focus on the Prp3 protein, which stably associates with the U4-U6 duplex, in combination with Snu13 and Prp31. They present a wealth of data on Prp3, combining classical biochemical approaches and mutant studies with protein- and RNA-protein crystallographic as well as phylogenetic data; all very carefully done, described and presented in great detail in high-quality figures. Their own work on the yeast complex (e.g. [46]) had shown that Prp3 contacts the U4-U6 duplex RNA on the U4-U6 stem II and U6-3' terminal regions. Here they extend these data by a combination of data, showing that the C-terminal region of Prp3 protein combines single- and double-stranded RNA-binding functions in corresponding regions: first, the DUF1115 domain at the very C-terminus of Prp3 recognizing the conserved single-stranded U6 3' terminus, preceded by, second, a protein domain that interacts with the double-stranded U4-U6 stem II.

Throughout this paper they compare yeast and human in both experiments and phylogenetic RNA/protein-sequence analyses, revealing many conserved features. They delineate the minimal domain at the C-terminal region of Prp3 protein required for stably binding an RNA of U4-U6 stem II and the U6 3' terminus (Figure 1: it may help to show schematically also the minimal U4-U6 RNA used in binding assays). This is used for solving the protein structure in that part of Prp3, in addition also together with a minimal U4-U6 RNA duplex. They are able to resolve the protein-RNA interaction at the U6-3' terminal region, but not at the U4-U6 stem region. RNA-protein contacts are probed and confirmed by mutant analysis. To investigate the protein recognition of the U4-U6 double-stranded region, they use hydroxyl radical footprinting with the minimal U4-U6 RNA fused into one RNA, which supports their model that the Prp3 peptide preceding the C-terminal DUF domain contacts the U4-U6 stem II.

They discuss their data in the context of the extensive mutant data from previous work in both yeast and human systems, in particular of the U6 snRNA mutant effects. Although earlier data were discussed mostly in terms of mutual stabilities of single-snRNA versus U4-U6 duplex RNAs, they add the aspect that these mutations likely also have effects on Prp3 protein binding. Overall, a very comprehensive and careful study I can recommend for publication.

Reviewer #3:

In the manuscript "A composite double-/single-stranded RNA-binding region in protein Prp3 supports tri-snRNP stability and splicing”, Liu et al. present a structural and functional analysis of the spliceosomal protein Prp3, which is implicated in reassembly of U4/U6 di-snRNA during recycling of the spliceosome. The manuscript is well written and mostly clear although sometimes not easy to follow due to complex nomenclature and description of residue and nucleotide numbers that are not always obvious. The results provide significant novel insight into our understanding of spliceosome assembly but also on protein-RNA recognition. Prp3 recognizes specifically an overhang of U6 snRNA, which is suggested to help to release Prp24 and enable binding to U4 snRNA in conjunction with other factors like Prp31 and Snu13.

The importance of the findings, quality of research, and presentation of results are in principle suitable for publication in *eLife*. However, there are some important aspects that need further attention and should be addressed by the authors:

The manuscript nicely and clearly demonstrates the recognition of the single-stranded RNA region by the globular domain in Prp3. However, the authors also report that the N-terminal region of their crystallization construct seems “to foster RNA contacts”, but that these are not discernible in the electron density. Later they add several experiments to support their initial findings about the importance of this region in U4/U6 di-snRNP assembly (maybe a reordering of the manuscript concerning this region could improve the reading flow). From the structure determined by X-ray crystallography it seems plausible that this presumably flexible tail is indeed present in the crystals and stabilizes the complex by contacts to the stem helix. Given that the manuscript mainly focuses on the structural biology of the complex, and that the recognition of the U4/U6 duplex region may also be an important aspect of the Prp3 interactions, it is a little disappointing that no additional efforts were made to further clarify the contribution of these regions for in complex formation. Straightforward experiments that would greatly help to clarify this point:

Binding affinities should be quantitatively determined comparing different constructs of the protein, i.e. with or without the disordered extension, but also using additional (point) mutations. Here, ITC, SPR or fluorescence polarization experiments could be employed.

Potentially, NMR binding studies could identify critical residues for the interaction in the unstructured region which could be further probed by mutational analysis.

---

## [Author Response]

*However, as you will see in the reviews below, there were also major differences of opinion regarding the work, including serious criticism. The hydroxyl radical footprinting results (*Figure 4*) were considered not very convincing, and should be replaced with another experiment or removed with a corresponding softening of the relevant conclusion. Regarding the U4-U6 stem-Prp3 interaction, further experimental details should be added, along the lines suggested by Reviewer 3, to strengthen this point*.

We agree and have replaced the footprinting experiment with other analyses. To test more rigorously whether the peptide preceding the DUF1115 domain is important for RNA binding and interaction with the stem II region, we now conducted additional binding experiments by EMSA titrations and ITC (as suggested by Reviewer 3). In both experimental setups, we find that wt Prp3 CTF binds to an RNA construct containing U4/U6 stem II and a 13 nucleotide U6 3’-overhang about two to three times more strongly than variants of the CTF bearing R304A or R322A exchanges in the N-terminal peptide or the DUF1115 domain alone. For details regarding these new data please refer to the new Figure panels 1C, 1D and 1F. We have also more carefully discussed this point in the beginning of the revised Discussion section.

*There was also some criticism on the presentation of the data, and both the Introduction, the first part of the Results, and some of the Discussion sections were considered difficult to read, fraught with too many details. Some of this is due to confusing switches between human and yeast nomenclature, and would require considerable rewriting and streamlining*.

We streamlined the entire manuscript.

Introduction:

We did not see specific criticism by the Reviewers concerning the Introduction. Regarding this section, we have followed the advice of Reviewer 3 and have included more original references in the first paragraph.

Results:

We agree that the detailed descriptions of our experiments with human and yeast proteins at the beginning of the Results section might have been confusing. We therefore have removed some details and have grouped the Results differently as suggested by Reviewer 3. We now start the Results with a short section on human Prp3, which we felt should remain in the manuscript, as we used this protein to experimentally define the C-terminal region as the major U4/U6 di-snRNA-binding portion of Prp3 orthologs. However, as mapping of the RNA structure requirements gave very similar results in the human and yeast systems, we removed the human RNA mapping experiments and now exclusively present these analyses for the yeast system. Also all following parts of the Results (additional biochemical, crystal structure and functional analyses) were conducted with yeast factors.

Furthermore, we have now combined all experiments, which relate to how portions of the protein and RNAs interact with each other, in the first part of the Results (i.e. quantification of relative affinities of different Prp3 fragments/variants and RNA variants as well as differential influence of Prp3 variants on Brr2-mediated U4/U6 unwinding). Finally, we have also placed the analysis of binding cooperativities among U4/U6 proteins directly after the first section on interaction mapping. In this fashion, all biochemical data on the minimal interacting regions and on the role of different protein portions in RNA binding and binding cooperativity are now grouped together at the beginning of the Results.

Discussion:

At the beginning of the revised Discussion, we now briefly summarize our experimental evidence that the C-terminal regions of Prp3 orthologs constitute a unique, conserved, composite ds/ss RNA-binding element. Following comments by Reviewer 1, we have now removed several of the more speculative parts of the Discussion (such as the possible role of disease [RP18]-related residues, for which we have no direct experimental evidence, as well as the possible role of Prp3 contacts to Prp24 and/or LSm proteins during U4/U6 re-assembly). We have retained a discussion of previous U4/U6 mutagenesis studies, as many of the previously probed residues are directly contacting Prp3 in our crystal structure; however, an influence of the manipulations on Prp3 binding was not considered at that time. Furthermore, we kept a brief discussion of possible mechanisms underlying the cooperative binding of U4/U6-interacting proteins, as we present data directly relating to this point. However, with respect to comments by Reviewer 1, we now briefly mention that the proteins might modulate each other’s interactions with the RNAs upon binding. Finally, although somewhat speculative, we have retained a discussion of a possible role of Prp3 in re-assembly of the U4/U6 di-snRNP. U6 snRNA regions contained in our crystal structures were also seen in a recently published Prp24-U6 snRNA complex structure, in which they take on a different conformation. Having these two structural snapshots in hand, we think it is important to comment on possible mechanisms of how one functional state (Prp24-U6) could interconvert into the other functional state (U4/U6 di-snRNP).

Reviewer #1:

*This manuscript presents crystallographic analysis of structures of Prp3 a U4/U6 di-snRNP. Structures of both Prp3 alone and bound to U4/U6 are presented. While some useful new information is obtained, e.g. what appears to be a novel RNA biding motif in Prp3, I have some concerns about the impact of the work. These include the fact that interaction between Prp3 and stem II of U4/U6 is not well resolved*.

Unfortunately, the relevant region of the protein is not well ordered in the crystal and thus did not allow us to resolve the presumed direct contacts of Prp3 to the stem II region of U4/U6. Although we tried extensively (also testing different RNA and protein constructs), we did not obtain a different crystal form, in which this region was better resolved. We now conducted additional EMSA titrations and ITC experiments using wt and mutant Prp3 constructs (lacking or containing the basic peptide preceding the DUF1115 domain and with or without point mutations in this preceding peptide) and various RNA constructs (new Figure panels 1C, 1D and 1F). The results are consistent with our idea that the peptide preceding the DUF1115 domain binds to the stem II portion of U4/U6 di-snRNAs. We have briefly summarized our combined experimental evidence regarding this point at the beginning of the revised Discussion.

*Additionally, the relevance of these structures in the absence of Snu13 and Prp31 is questionable given that the three proteins bind U4/U6 cooperatively. It seems quite possible that the presence of Snu13 and or Prp31 could alter the functionally relevant conformation of Prp3*.

While it is certainly possible that Prp3, Prp31 and Snu13 contact each other on U4/U6, we consider it highly unlikely that such interactions would change the conformation of the DUF1115 domain and the way it interacts with the U6 3’-overhang, which constitutes a major aspect of our work. However, it is imaginable that the N-terminal stem II-binding peptide might be influenced by the other two proteins and we therefore now briefly point this out in the revised Discussion:

“It is also conceivable that the Prp3 stem II-binding peptide directly contacts Snu13 and/or Prp31 on the RNAs and that the latter two proteins even modulate how the peptide interacts with stem II.”

Reviewer #2:

*Throughout this paper they compare yeast and human in both experiments and phylogenetic RNA/protein-sequence analyses, revealing many conserved features. They delineate the minimal domain at the C-terminal region of Prp3 protein required for stably binding an RNA of U4-U6 stem II and the U6 3' terminus (*Figure 1*: it may help to show schematically also the minimal U4-U6 RNA used in binding assays). This is used for solving the protein structure in that part of Prp3, in addition also together with a minimal U4-U6 RNA duplex. They are able to resolve the protein-RNA interaction at the U6-3' terminal region, but not at the U4-U6 stem region. RNA-protein contacts are probed and confirmed by mutant analysis. To investigate the protein recognition of the U4-U6 double-stranded region, they use hydroxyl radical footprinting with the minimal U4-U6 RNA fused into one RNA, which supports their model that the Prp3 peptide preceding the C-terminal DUF domain contacts the U4-U6 stem II*.

We agree and have now included schemes of RNA constructs used in all relevant Figures of the revised manuscript.

Reviewer #3:

*The manuscript nicely and clearly demonstrates the recognition of the single-stranded RNA region by the globular domain in Prp3. However, the authors also report that the N-terminal region of their crystallization construct seems “to foster RNA contacts”, but that these are not discernible in the electron density. Later they add several experiments to support their initial findings about the importance of this region in U4/U6 di-snRNP assembly (maybe a reordering of the manuscript concerning this region could improve the reading flow)*.

We have restructured the manuscript as suggested. In the revised manuscript, all biochemical data on the minimal interacting regions and on the role of different protein portions in RNA binding and U4/U6 unwinding by Brr2 are now grouped together at the beginning of the Results. This is followed by a brief section on the binding cooperativity with other U4/U6 proteins.

*From the structure determined by X-ray crystallography it seems plausible that this presumably flexible tail is indeed present in the crystals and stabilizes the complex by contacts to the stem helix. Given that the manuscript mainly focuses on the structural biology of the complex, and that the recognition of the U4/U6 duplex region may also be an important aspect of the Prp3 interactions, it is a little disappointing that no additional efforts were made to further clarify the contribution of these regions for in complex formation. Straightforward experiments that would greatly help to clarify this point*:

*Binding affinities should be quantitatively determined comparing different constructs of the protein, i.e. with and without the disordered extension, but also using additional (point) mutations. Here, ITC, SPR or fluorescence polarization experiments could be employed*.

We agree with the reviewer and have conducted additional EMSA titrations and ITC experiments to quantify the relative affinities of the wt Prp3 CTF (DUF1115 domain plus preceding basic peptide), the same fragment bearing single Arg-to-Ala exchanges in the N-terminal peptide presumed to bind stem II (R304A and R322A) as well as of the DUF1115 domain alone (without the preceding peptide) to an RNA construct containing U4/U6 stem II and a 13 nucleotide U6 3’-overhang. The trends in the affinities (wt binding several fold more strongly than R304A, R322A or DUF1115) support the idea that the peptide preceding the DUF1115 domain also elicits RNA contacts. We also carried out additional EMSA titrations with RNA constructs, in which we converted a conserved non-canonical pyrimidine-pyrimidine base pair of stem II to a Watson-Crick pair. For details on the quantifications of the relative interactions strengths please refer to the new Figure panels 1C, 1D and 1F.

Together, the results suggest that there are contacts of the C-terminal fragment of Prp3 to stem II. As based on our crystal structures, contacts to stem II cannot originate from the DUF1115 domain, our results suggest that the preceding basic peptide entertains these interactions with stem II. Our arguments concerning the Prp3-U4/U6 interaction mode are now briefly summarized at the beginning of the new Discussion section.

*Potentially, NMR binding studies could identify critical residues for the interaction in the unstructured region which could be further probed by mutational analysis*.

Simple NMR-based binding studies would only yield similar insight as, e.g., ITC (i.e. allow to assess differences in affinities of different protein variants), which we now have conducted. An in-depth NMR study to identify residues that directly contact the RNAs would not only require the production of more complex isotope-labeled proteins but also lengthy resonance assignments, which are out of the scope of the present work and not part of our own expertise.